# Irregular spiking of pyramidal neurons organizes as scale-invariant neuronal avalanches in the awake state

**Timothy Bellay[†], Andreas Klaus[†], Saurav Seshadri, Dietmar Plenz\***

Section on Critical Brain Dynamics, National Institute of Mental Health, Bethesda, United States

**Abstract** Spontaneous fluctuations in neuronal activity emerge at many spatial and temporal scales in cortex. Population measures found these fluctuations to organize as scale-invariant neuronal avalanches, suggesting cortical dynamics to be critical. Macroscopic dynamics, though, depend on physiological states and are ambiguous as to their cellular composition, spatiotemporal origin, and contributions from synaptic input or action potential (AP) output. Here, we study spontaneous firing in pyramidal neurons (PNs) from rat superficial cortical layers in vivo and in vitro using 2-photon imaging. As the animal transitions from the anesthetized to awake state, spontaneous single neuron firing increases in irregularity and assembles into scale-invariant avalanches at the group level. In vitro spike avalanches emerged naturally yet required balanced excitation and inhibition. This demonstrates that neuronal avalanches are linked to the global physiological state of wakefulness and that cortical resting activity organizes as avalanches from firing of local PN groups to global population activity.

**\*For correspondence:** plenzd@
mail.nih.gov

[†]These authors contributed
equally to this work

**Competing interests:** The
authors declare that no
competing interests exist.

**Reviewing editor**: Frances K
Skinner, University Health
Network, and University of
Toronto, Canada

## Introduction

When the brain is not engaged in any particular sensory, cognitive, or motor task, cortical neurons nevertheless give rise to coordinated group activity. This so-called resting activity delineates functional networks (*Fox and Raichle, 2007*; *Haimovici et al., 2013*), modulates responses to sensory input (*Arieli et al., 1996*; *Womelsdorf et al., 2012*), predicts cortical responses (*Luczak et al., 2009*), and changes in disease or sleep deprived states (*Greicius et al., 2004*; *Meisel et al., 2013*). It is, therefore, crucial to identify and understand the dynamical constraints that determine coordinated neuronal group activity at rest in the awake (AW) state.

Resting activity, recorded from neuronal populations in vivo using the local field potential (LFP; *Gireesh and Plenz, 2008*; *Petermann et al., 2009*; *Yu et al., 2011*; *Priesemann et al., 2013*), functional magnetic resonance imaging (fMRI; *Fraiman and Chialvo, 2012*; *Haimovici et al., 2013*), the magnetoencephalogram (MEG; *Palva et al., 2013*; *Shriki et al., 2013*), or electrocorticogram (ECoG; *Solovey et al., 2012*), has been shown in rodents, non-human primates, and humans to be composed of activity cascades called neuronal avalanches. Neuronal avalanches identify a specific organization of activity patterns in which avalanche sizes distribute according to a power law with slope of −1.5, that is, the relative occurrences of avalanche sizes are constant (*Beggs and Plenz, 2003*). This scale-invariant organization, which transcends many spatial and temporal scales, is an indication that cortical dynamics reside at or close to a critical state in which interactions between local elements give rise to long-range spatial and long-term temporal fluctuations (*Plenz and Thiagarajan, 2007*; *Chialvo, 2010*; *Beggs and Timme, 2012*; *Plenz, 2012*; *Marković and Gros, 2014*; *Plenz and Niebur, 2014*). Predictions from the theory of criticality (*Bertschinger and Natschlager, 2004*; *Nykter et al., 2008*; *Chialvo, 2010*; *Plenz and Niebur, 2014*) and neural modeling

**eLife digest** Even when we are not engaged in any specific task, the brain shows coordinated patterns of spontaneous activity that can be monitored using electrodes placed on the scalp. This resting activity shapes the way that the brain responds to subsequent stimuli. Changes in resting activity patterns are seen in various neurological and psychiatric disorders, as well as in healthy individuals following sleep deprivation.

The brain's outer layer is known as the cortex. On a large scale, when monitoring many thousands of neurons, resting activity in the cortex demonstrates propagation in the brain in an organized manner. Specifically, resting activity was found to organize as so-called neuronal avalanches, in which large bursts of neuronal activity are grouped with medium-sized and smaller bursts in a very characteristic order. In fact, the sizes of these bursts—that is, the number of neurons that fire—are found to be scale-invariant, that is, the ratio of large bursts to medium-sized bursts is the same as that of medium-sized to small bursts. Such scale-invariance suggests that neuronal bursts are not independent of one another. However, it is largely unclear how neuronal avalanches arise from individual neurons, which fire simply in a noisy, irregular manner.

Bellay, Klaus et al. have now provided insights into this process by examining patterns of firing of a particular type of neuron—known as a pyramidal cell—in the cortex of rats as they recover from anesthesia. As the animals awaken, the firing of individual pyramidal cells in the cortex becomes even more irregular than under anesthesia. However, by considering the activity of a group of these neurons, Bellay, Klaus et al. realized that it is this more irregular firing that gives rise to neuronal avalanches, and that this occurs only when the animals are awake. Further experiments on individual pyramidal cells grown in the laboratory confirmed that neuronal avalanches emerge spontaneously from the irregular firing of individual neurons. These avalanches depend on there being a balance between two types of activity among the cells: 'excitatory' activity that causes other neurons to fire, and 'inhibitory' activity that prevents neuronal firing.

Given that resting activity influences the brain's responses to the outside world, the origins of neuronal avalanches are likely to provide clues about the way the brain processes information. Future experiments should also examine the possibility that the emergence of neuronal avalanches marks the transition from unconsciousness to wakefulness within the brain.

(*Beggs and Plenz, 2003*; *Kinouchi and Copelli, 2006*; *Rämö et al., 2007*; *Shew et al., 2009*; *Tanaka et al., 2009*; *de Arcangelis and Herrmann, 2010*) suggest that cortical networks that reside in such a fluctuation-dominated regime can improve various aspects of information processing, as was demonstrated experimentally in vitro (*Shew et al., 2009*, *2011*; *Yang et al., 2012*; *Shew and Plenz, 2012*). Yet, further exploration of the origin and potential functional advantages of avalanche dynamics are limited by the ambiguity in the composition of the LFP, ECoG, MEG, and BOLD fMRI signals, with respect to their spatiotemporal and cellular origin as well as synaptic input and action potential (AP) output.

It is now well established that neuronal avalanches can emerge within cortex as demonstrated in vitro (*Beggs and Plenz, 2003*; *Stewart and Plenz, 2006*; *Pasquale et al., 2008*), and sensibly depend on the balance of fast synaptic excitation/inhibition (E/I) (*Beggs and Plenz, 2003*; *Pasquale et al., 2008*; *Shew et al., 2009*) and neuromodulators (*Stewart and Plenz, 2006*; *Pasquale et al., 2008*). However, it is currently not known how the scale-invariant, macroscopic organization of avalanches measured at the population level relates to the output of the principal cells of cortex, that is, AP firing in pyramidal neurons (PNs) and how this organization relates to the global physiological state of the animal. AP firing in PNs is commonly reported as spontaneous and irregular (*Softky and Koch, 1993*; *Shadlen and Newsome, 1998*) with low average correlation in firing between neurons during spontaneous activity and a high variability in evoked AP responses (*Gawne and Richmond, 1993*; *Kerr et al., 2007*; *Sato et al., 2007*; *Poulet and Petersen, 2008*; *Ecker et al., 2010*; *Komiyama et al., 2010*; *Renart et al., 2010*). Here, we show experimentally in vivo that ongoing fluctuations in AP firing in single cortical neurons amount to scale-invariant AP avalanches at the neuronal group level. The emergence of spike avalanches marks the AW state and is absent under anesthesia. Similarly, spike avalanches spontaneously organize in layer 2/3 PN groups from

organotypic slice cultures and yet are sensitive to the E/I balance. We propose that critical dynamics govern the organization of resting activity in the awake animal from AP firing in individual PNs to the activity in large neuronal groups across cortex.

## Results

### Highest variability in firing in awake resting

To identify the relationship between AP firing and neuronal avalanches, which are primarily found in superficial layers of cortex (*Stewart and Plenz, 2006*; *Petermann et al., 2009*), we expressed the genetically encoded calcium indicator (GECI) YC2.60 (*Yamada et al., 2011*) in layer 2/3 (L2/3) PNs of rats using in utero electroporation at embryonic day $15.5 \pm 0.5$ (*Saito, 2006*). Labeled mature neurons distributed throughout dorsolateral frontal and sensorimotor cortex. They exhibited PN morphology (*Figure 1A*), and their synaptic transmission was blocked by glutamate receptor antagonists (data not shown). To record ongoing AP activity in local PN groups, we performed 2-photon imaging (2-PI) of YC2.60-expressing PNs in head-restrained rats (*Figure 1A,B*; depth = $270 \pm 50$ μm; cortical area = $0.15 \pm 0.05$ mm$^2$; 10–15 min per recording). Recordings were done under anesthesia (AN; 1–2% isoflurane), during wakening (WK; 5–20 min at 0% isoflurane), and in the awake state (AW; after >20 min at 0% isoflurane). Intracellular calcium transients produced fluorescence changes in visually identified somatic ROIs (*Figure 1B*), which were converted into ratiometric time courses ($\Delta R/R$) and then deconvolved to obtain an instantaneous firing rate estimate, $\lambda$, for each neuron (*Vogelstein et al., 2010*) (see 'Materials and methods'; *Figure 1C,D*). In control experiments, we showed (1) YC2.60 reliably and linearly reported AP activity at physiological temperature from single APs to AP bursts up to 28 Hz (*Figure 1—figure supplement 1A,B*) and (2) $\lambda$ linearly recovered spike trains at different temporal resolutions (*Figure 1—figure supplement 2*). We first recorded at a temporal resolution of $\Delta t = 250$ ms (n = 6 rats; 38 recordings; 15–30 active PNs/recording; >1 AP/min). Neuronal activity was stationary in $\lambda$ and in the average crosscorrelation, R, between ROIs (*Figure 1E,F*, respectively). Neurons fired on average more during AW compared to WK and AN (ANOVA, $F(2,35) = 23.05$, $p < 0.001$; probability density function (PDF) shown in *Figure 1G*). Under all three conditions, though, neurons fired irregular APs interspaced by relatively long periods of quiescence. This was quantified by three measures. First, $\lambda$ distributed exponentially for single neurons [log-likelihood ratio (LLR) comparison between power law vs exponential, >98% of ROIs in favor of exponential distribution, $p < 0.05$; *Figure 2A*, single distributions and average for one PN group; *Figure 2B*, averages over all PN groups]. Second, neurons tended to not fire at all within $\Delta t$ (*Figure 2A,B*, *left*; *arrow*). The corresponding probability of quiescence, $P_q$ ($\lambda$ < minimal $\lambda$ threshold, $\lambda_{thr}$, set to 0.5), was highest for AN and WK (*Figure 2B*, *inset*; ANOVA, $F(2,35) = 23.05$, $p = 0.002$). Both of these characteristics remained true for higher temporal resolutions despite the expected increase in $\lambda$ fluctuations (*Figure 2B*, *right*; additional n = 6 rats; n = 19 recordings; $\Delta t = 167$ and 88 ms during AW; LLR in favor of exponential, $p < 0.05$) and $P_q$ (ANOVA, $F(2,28) = 31.46$, $p < 0.001$). Third, the normalized duration of quiescent times, $IBI_{norm}$, between firing (i.e., $\lambda < \lambda_{thr} = 0.5$) also distributed exponentially for all conditions (*Figure 2A*, *right*; *Figure 2C*; LLR: >98% of ROIs with $p < 0.05$). The corresponding CV was larger than 1 for all conditions, was significantly higher for AW than WK and AN (*Figure 2C*, *left*; *inset*, AW: $1.5 \pm 0.2$; WK: $1.2 \pm 0.1$; AN: $1.2 \pm 0.1$; mean $\pm$ SD, $F(2,35) = 25.66$, $p < 0.001$), and increased further with temporal resolution (AW, *Figure 2C*, *right*; $\Delta t = 167$ ms, $1.9 \pm 0.4$; $\Delta t = 88$ ms, $2.1 \pm 0.3$, mean $\pm$ SD; $F(2,28) = 15.48$, $p < 0.001$). This irregularity was also robust to minimal AP activity: increasing $\lambda_{thr}$ smoothly reduced the average firing rate $\lambda_{avg}$ (data not shown), yet maintained a CV larger than 1 for all conditions and $\Delta t$ (*Figure 2—figure supplement 1*). CV values for single units (n = 26; average firing rate = 1.3 Hz; range: 0.1–6.2 Hz) recorded with chronic microelectrode arrays from superficial layers in the AW rat (*Figure 1—figure supplement 2*) compared favorably with $\lambda$ results from our imaging analysis and ranged between $1.6 \pm 0.4$ ($\Delta t = 0.033$ ms) and $1.3 \pm 0.3$ (resampled at $\Delta t = 250$ ms), respectively.

To further differentiate the high irregularity encountered in the AW resting state from WK and AN conditions, we studied the temporal and spatial clustering of PN firing. Previous work on the AW resting state revealed temporal AP clustering during large intracellular membrane potential fluctuations (*Poulet and Petersen, 2008*) and spatial clustering of AP firing for L2/3 PNs (*Greenberg et al., 2008*) similar to the correlation profile found during tasks (*Komiyama et al., 2010*). Indeed, we

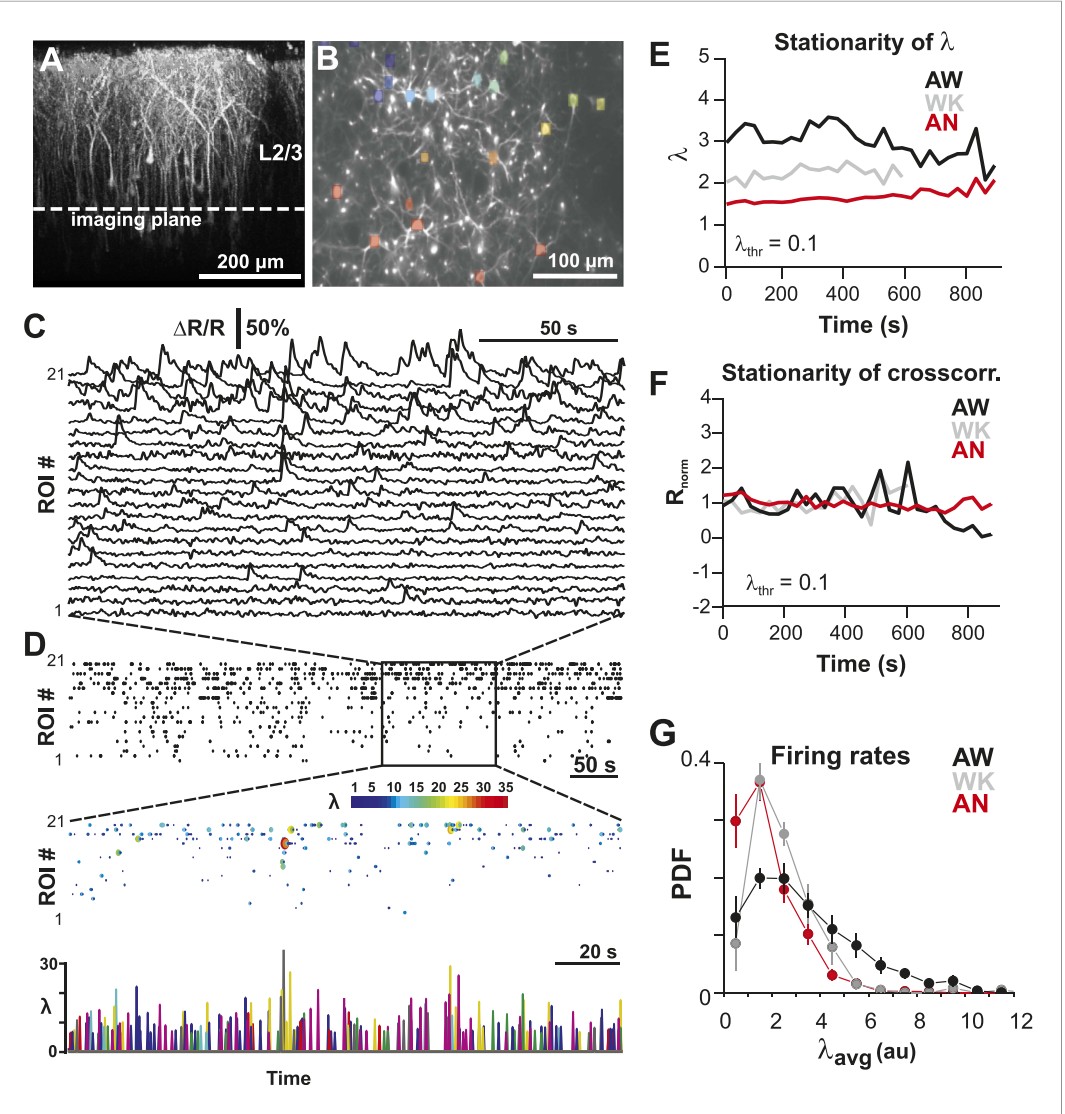

**Figure 1**. Imaging of ongoing spiking activity in groups of L2/3 PNs in the awake (AW) rat. (**A**) Z-stack side projection of YC2.60-expressing PNs in L2/3 in vivo. (**B**) Single imaging plane (dotted line in **A**) containing a group of PNs with significant changes in fluorescent intensity ΔR/R over time (colored ROIs). (**C**) Time course of ΔR/R for individual ROIs (from **B**). (**D**) *Top*: Binary raster display of instantaneous spike rate estimate λ ($\lambda_{thr} = 0.1$). *Middle*: Expanded period showing color coded λ amplitude. *Bottom*: Overplot of λ time course for individual, color-coded ROIs. (**E**, **F**) Stationary firing rate estimate $\lambda$ and pairwise crosscorrelation $R_{norm}$ (normalized by the correlation during the first 30 s) over the course of acquisition. Firing rate increased from anesthetized to AW conditions but remained stable throughout the recording, suggesting that our measures were not affected by slow modulations of activity (*Ecker et al., 2010*). Shown are averages over all PN groups. (**G**) Distributions of the average firing rate estimate, $\lambda_{avg}$, for the three different states.

The following figure supplements are available for figure 1:

**Figure supplement 1**. Single AP detection in YC2.60-expressing neurons at physiological temperature and performance of the OOPSI deconvolution algorithm.

**Figure supplement 2**. Performance of the OOPSI deconvolution algorithm at different temporal resolutions and noise levels.

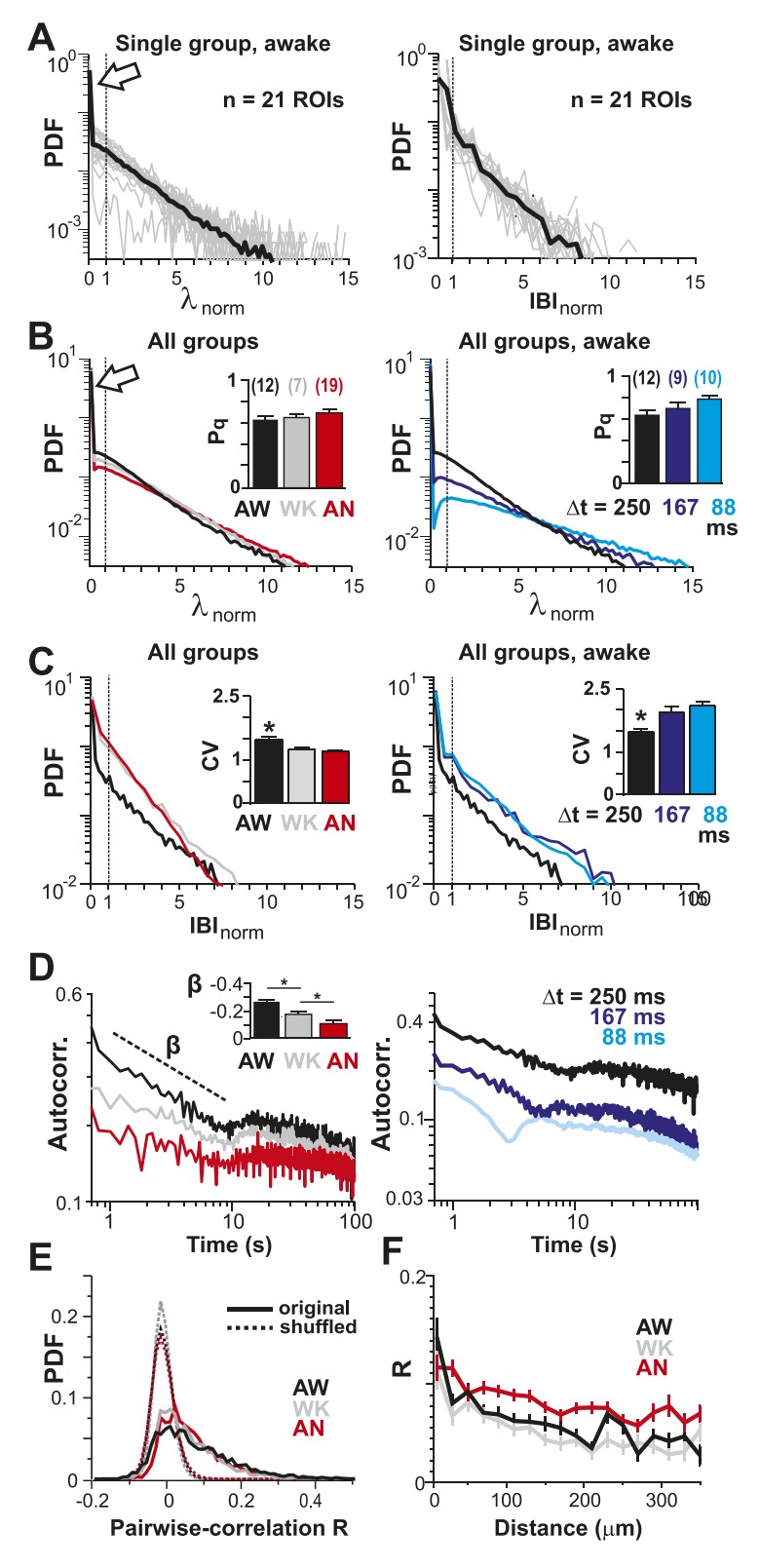

**Figure 2**. Spatial and temporal clustering in ongoing spiking activity in vivo. (**A**) Probability distributions of $\lambda_{norm} = \lambda/\lambda_{avg}$ (*left*) and distribution of normalized quiescent time intervals, $IBI_{norm} = IBI/IBI_{avg}$, (*right*) in the AW state for a single neuronal group of PNs ($\Delta t = 250$ ms). *Gray*: distributions for individual ROIs, *black*: average. Dotted lines, $\lambda_{norm} = 1$ and $IBI_{norm} = 1$. The arrow is pointing at the relatively high-probability

*Figure 2. Continued*

function value for $\lambda_{norm} \ll 1$, that is, no spike within $\Delta t = 250$ ms. Note that the transition from 'no spike' to spiking is rather abrupt in the distribution, which indicates the high signal-to-noise ratio in our recorded data (*cf.* **Figure 1—figure supplement 1A,B**). (**B**) Average $\lambda_{norm}$ distributions over all PN groups for all three conditions (*left*; AW; WK, waking; AN, anesthetized) and temporal resolutions (*right*). *Inset*: probability of quiescence $P_q$ (number of recordings is indicated in parenthesis). (**C**) Distribution of $IBI_{norm}$ for all three conditions (*left*) and temporal resolutions (*right*). *Inset*: Coefficient of variation (CV) for IBI (*p < 0.05). (**D**) Average autocorrelation function for $\lambda$ across PNs for all three conditions (*left*; $\Delta t = 250$ ms) and temporal resolutions (*right*). *Inset*: Power-law exponent $\beta$ (*p < 0.05). Note steeper power-law decay for AW indicating increased temporal clustering. (**E**) Distribution of pairwise crosscorrelation, *R*, in $\lambda$ for all PN groups and different states. *Broken lines*: Corresponding independent model by shuffling $\lambda$ for each ROI. (**F**) AW shows steeper distance decay in *R* when compared to AN indicating higher spatial clustering.

The following figure supplement is available for figure 2:

**Figure supplement 1**. CV in $\lambda$ remains larger than one for increasing $\lambda_{thr}$ for all conditions (**A**) and temporal resolutions (**B**).

found that $\lambda$ was more correlated in time during AW compared to WK and AN, as demonstrated by a significantly steeper decay in the autocorrelation for periods <10 s (**Figure 2D**, *left*; ANOVA, $F(2,35) = 14.77$, $p < 0.001$). On the other hand, correlated firing between pairs of neurons was weak, in line with the notion of an 'asynchronous state of cortex' (**Poulet and Petersen, 2008**; **Ecker et al., 2010**; **Renart et al., 2010**), and the average did not differ between states (AW: $0.06 \pm 0.06$; WK: $0.05 \pm 0.04$; AN: $0.06 \pm 0.04$, mean $\pm$ SD; ANOVA, $F(2,35) = 0.29$, $p = 0.75$; **Figure 2E**). While neighboring neurons tended to be correlated more than distant neurons, in line with previous reports (**Sato et al., 2007**; **Greenberg et al., 2008**), this spatial profile was largely similar across all three states (**Figure 2F**).

## Avalanches emerge from awake state neuronal firing

In the preceding section, we quantified how irregular spontaneous firing in individual PNs and their pairwise correlations change as the animal transitions from the AN to the AW state. None of these measures, though, allows us to identify neuronal avalanches, which reflect a scale-invariant relationship of neuronal group activities. In fact, we recently demonstrated that event rate and pairwise correlation R are insufficient to predict neuronal avalanches in cortical activity (**Yu et al., 2011**). In a first approach, we therefore identified spatiotemporal activity clusters in the neuronal population. This was done by concatenating firing events of neurons that co-occurred either within $\Delta t$ or within consecutive periods of $\Delta t$ (**Figure 3A**; *gray areas*) and separating clusters by quiescent periods of at least $\Delta t$, the original approach to identify avalanche dynamics (**Beggs and Plenz, 2003**). For a given neuronal population and 2-PI, this process has two free parameters: (1) the temporal resolution $\Delta t$, which is fixed by the scanning frame rate of 2-PI and (2) the activity threshold, $\lambda_{thr}$, of a firing event. In general, if $\lambda_{thr}$ is low, most firing events will be concatenated into few large clusters. Similarly, if $\lambda_{thr}$ is high, the few remaining firing events will group into few clusters. Thus, a maximal number of clusters is expected at an intermediate threshold $\lambda_{thr}^{max}$. We first studied this relationship in the AW state. Indeed, for a given recording at $\Delta t$, the cluster rate increased with $\lambda_{thr}$ and was maximal at an intermediate threshold $\lambda_{thr}^{max}$ (**Figure 3B**, *arrows*). As expected, $\lambda_{thr}^{max}$ shifted towards smaller $\lambda_{thr}$ values at higher temporal resolutions due to the improved resolution of fast $\lambda$ fluctuations. Next, we studied the cluster size *s*, that is, the sum of all firing events within a cluster normalized by the predicted cluster size limit $\Lambda$, which is determined by the number of ROIs and their respective average firing rate (see 'Materials and methods'). If the activity of neurons was rather independent from each other, as one might assume from the low average pairwise correlation in $\lambda$ (**Figure 1**), the distribution in cluster size should be close to an exponential function. On the other hand, if interactions between neurons contribute significantly to spontaneous firing, then the cluster size distribution deviates from an exponential function, and, in the case of avalanche dynamics, should follow a power law (**Plenz and Thiagarajan, 2007**). Importantly, we found that cluster sizes distributed according to a power law over approximately two orders of magnitude (**Figure 3C**: $\alpha = 1.63 \pm 0.13$,

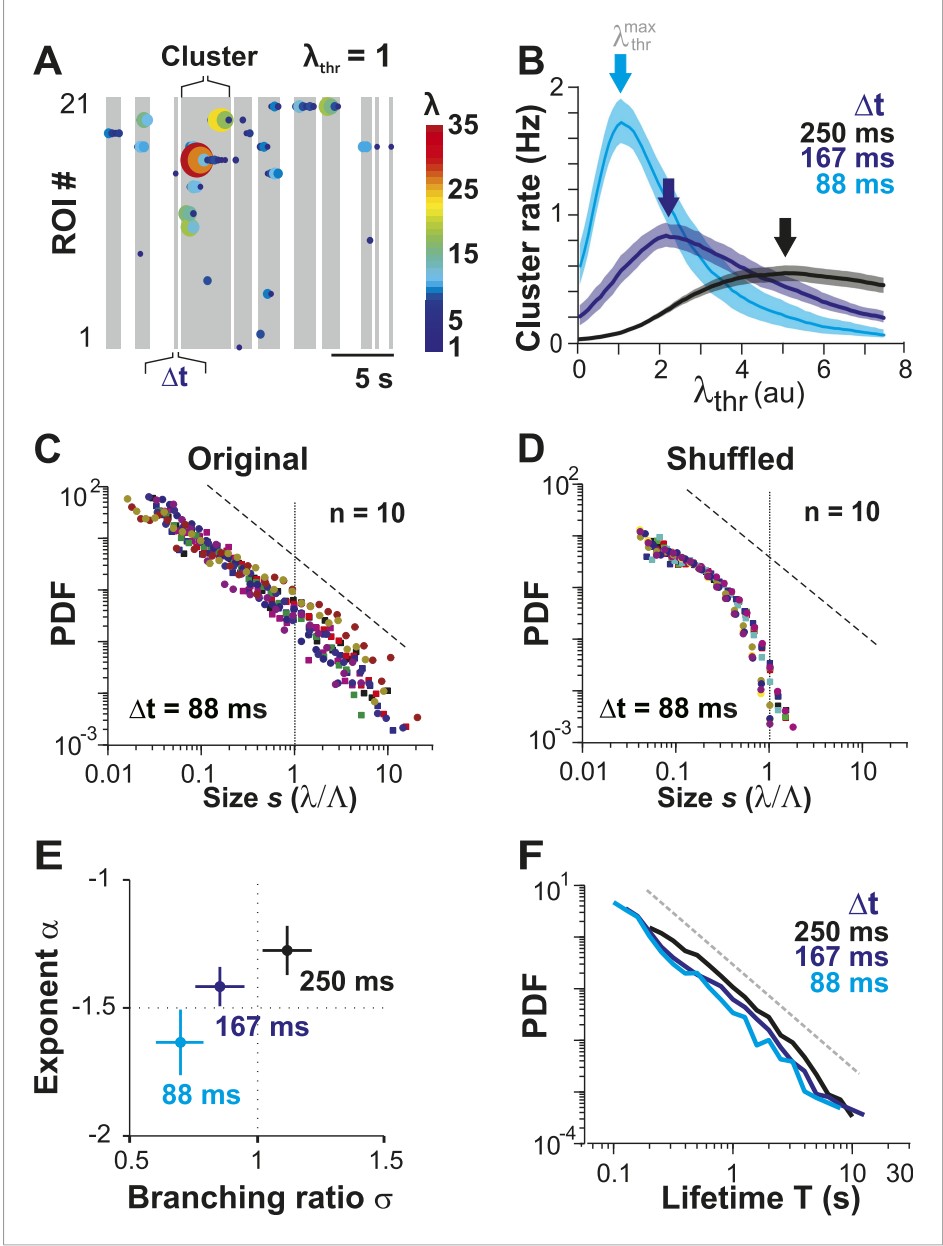

**Figure 3.** Ongoing spiking in local PNs organizes as neuronal avalanches in vivo. (**A**) Sketch of cluster formation at given Δt and chosen $\lambda_{thr} = 1$. Gray boxes delineate clusters of activity (i.e., consecutive time bins with $\lambda > \lambda_{thr}$). (**B**) Maximal cluster rate at intermediate $\lambda_{thr}$ for different Δt in the AW condition. Vertical arrows indicate the respective $\lambda_{thr} = \lambda_{thr}^{max}$ at which cluster rate is maximal. (**C**) Individual distributions of normalized cluster sizes, s, in AW (top; Δt = 88 ms, n = 10 recordings, threshold at $\lambda_{thr}^{max}$). Dotted line, predicted cut-off at s = 1; dashed line, power law with α = −1.5. (**D**) Corresponding distributions after shuffling λ. Shuffling destroys spatiotemporal correlations in activity and abolishes the power law in cluster sizes. (**E**) Relationship between α and branching ratio σ for all three temporal resolutions, Δt. Note the systematic change for increasing Δt as shown previously for avalanche dynamics based on the LFP. (**F**) Distribution of cluster lifetimes, T, for different Δt. Dashed line, slope = −2.

LLR = 25.7–201.4 favors power law over exponential, p < 0.003 for all n = 10 experiments, Δt = 88 ms at individual $\lambda_{thr}^{max}$). To determine whether the AP activities in PN groups that resulted in power-law distributed cluster sizes were indeed a result of spatiotemporal correlations, we performed, as a control, time-shuffling of the corresponding λ events. As shown in **Figure 3D**, time-shuffled λ events did not yield power-law size distributions, and instead, cluster size distributions were better fit by an

exponential (LLR = −66.6 to −6.9, favors exponential over power law, p < 0.05 for 7/10 experiments, Δt = 88 ms; thresholded at $\lambda_{thr}^{max}$ obtained for each distribution individually).

For LFP-based avalanche dynamics, it has consistently been shown that power-law exponent and branching ratio increase systematically with temporal resolution Δt (*Beggs and Plenz, 2003*; *Petermann et al., 2009*). Importantly, for critical avalanche dynamics and negligible finite-size effects, the temporal resolution for which the power-law exponent, α, is −1.5 yields a branching ratio, σ, close to 1. As shown in *Figure 3E*, a similar relationship between α and σ was also found empirically for AP avalanches in vivo. Furthermore, the temporal organization of neuronal avalanches, that is, the avalanche life time, was shown to distribute according to a power law with exponent close to −2 (experimentally: [*Beggs and Plenz, 2003*]; simulations and theory: [*Harris, 1989*; *Eurich et al., 2002*]). Similarly, we found that the cluster lifetime, T, distributed according to a power law with slope close to −2 (*Figure 3F*; Δt = 250 ms, slope γ = 1.7 ± 0.1; Δt = 167 ms, γ = 1.9 ± 0.2; Δt = 88 ms, γ = 2.2 ± 0.2, LLR = 100.4–344.1; p < 0.001). To study the robustness of the power-law size distributions with respect to threshold, we systematically varied $\lambda_{thr}$ around $\lambda_{thr}^{max}$. In *Figure 4*, we show that the body of the distributions followed a power law up to the predicted cluster size limit (s = λ/Λ = 1 for the normalized distributions) beyond which a cut-off was observed. This cut-off was more pronounced at lower temporal resolutions and higher thresholds as shown previously for LFP-based avalanches (*Yu et al., 2014*). The threshold robustness and cut-off are in line with previous reports on avalanche dynamics in vitro (*Beggs and Plenz, 2003*) and in non-human primates (*Petermann et al., 2009*) and humans (*Shriki et al., 2013*).

Avalanche dynamics were unique to the AW state (*Figure 5*). During AN, cluster size distributions at corresponding $\lambda_{thr}^{max}$ (*Figure 5—figure supplement 1A*) were slightly bimodal (*Figure 5A*, *arrow*), in line with a progressively worse fit to a power law for WK and AN compared to AW (*Figure 5B*; ANOVA, p < 0.05; *cf. Figure 4*, *Figure 5—figure supplement 1B*).

An alternative approach to obtain avalanches, in which periods of integrated population activity above a population threshold were extracted (*Poil et al., 2012*), also yielded power-law size distributions with exponent close to −1.5 and cut-off that were robust to changes in $\lambda_{thr}$ (*Figure 6A–D*). Similar to what was found when using the original definition of avalanches, cluster size distributions obtained by population thresholding deviated from avalanche dynamics under isoflurane anesthesia (*Figure 5B*, *Figure 6E*, p < 0.01; Kruskal–Wallis test on Kolmogorov–Smirnov distances, $D_{KS}$). Furthermore, cluster size and lifetime were correlated, and the corresponding exponent scaled as suggested by the theory of critical systems (*Sethna et al., 2001*) (*Figure 6—figure supplement 1*).

To summarize, ongoing AP firing of local groups of L2/3 PNs in the AW state displayed the five hallmarks of neuronal avalanche dynamics: first, a power law in size distributions with exponent close to −1.5; second, a critical branching parameter close to 1; third, threshold robustness; fourth, a lifetime distribution with exponent close to −2; and fifth, scaling of lifetimes and sizes. These results, for the first time, demonstrate the emergence of neuronal avalanches in the spiking of PN groups from superficial cortical layers in the AW animal.

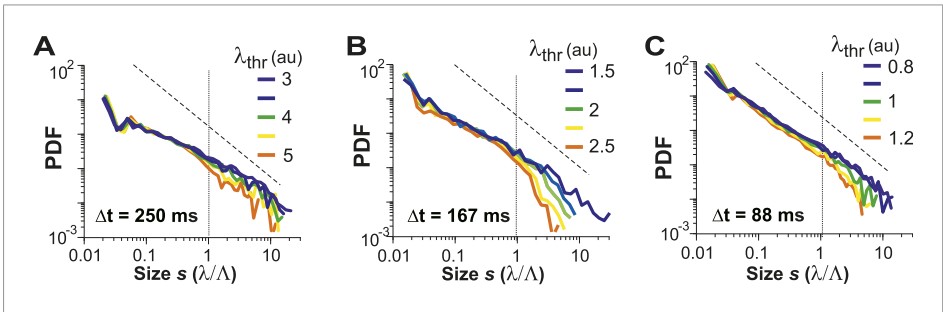

**Figure 4**. Avalanche dynamics is robust to changes in $\lambda_{thr}$. (**A–C**) Cluster size distributions for Δt = 250, 167, and 88 ms (from *left* to *right*). The green distributions correspond to the respective thresholds, $\lambda_{thr} = \lambda_{thr}^{max}$, at which the cluster rate was maximal.

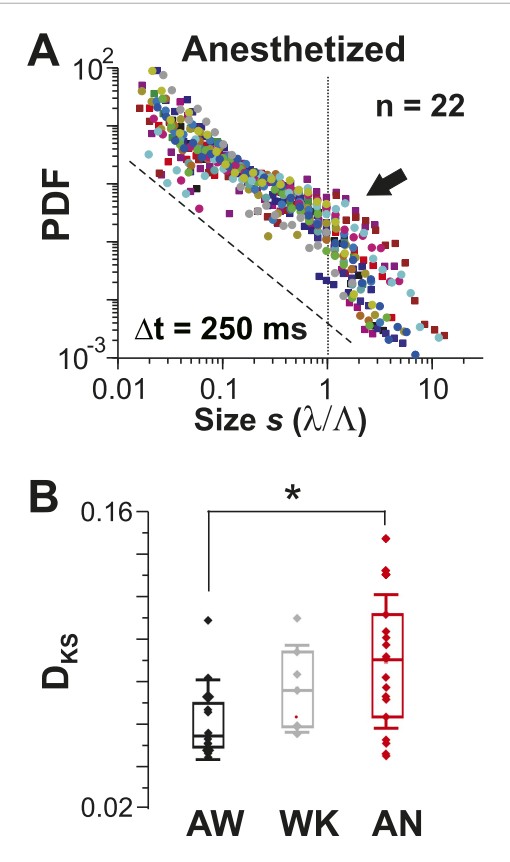

**Figure 5**. Avalanche dynamics is abolished under anesthesia. (**A**) Overplot of size distributions for the anesthetized state (*n* = 22 recordings) showing a slight increase in the probability of large clusters (*arrow*). (**B**) Average KS distance, $D_{KS}$, between individual PDFs and best-fit power-law distributions for all three states (*p < 0.05).

The following figure supplement is available for figure 5:

**Figure supplement 1**. (**A**) Maximum cluster rate is observed at intermediate threshold levels for all three conditions.

## AP avalanches in L2/3 PNs in vitro depend on the E/I balance

LFP recordings in cortex slice cultures (*Beggs and Plenz, 2003*; *Stewart and Plenz, 2007*; *Gireesh and Plenz, 2008*) have shown avalanche dynamics to emerge spontaneously in superficial layers. Similarly, spike avalanches have been identified in extracellular unit recordings from dissociated cultures of hippocampus (*Mazzoni et al., 2007*) and cortex (*Pasquale et al., 2008*; *Tetzlaff et al., 2010*; *Vincent et al., 2012*), although the mesoscopic organization of the tissue was not preserved. Yet, these studies are limited by the unknown composition of the LFP population signal (see 'Introduction') and cell types recorded from. For extracellular unit activity, strongly bursting interneurons can dominate large spike clusters in the neuronal population, in which case heavy-tailed cluster size distributions reflect neuronal differences rather than neuronal interactions. In order to demonstrate that avalanche dynamics also capture spatiotemporal activity in L2/3 PNs in vitro, we conducted studies in GECI-expressing cortical slices, co-cultured with VTA to ensure proper maturation of superficial cortical layers (*Gireesh and Plenz, 2008*) (*Figure 7A–D*). We recorded AP activity from local groups of L2/3 PNs in vitro (*n* = 15–80 ROIs) monitored with YC2.60 (Δt = 250 ms; *n* = 129 movies, *n* = 35 cultures; *Figure 7B–D*) and compared the activity to conditions when $GABA_A$ (5 μM PTX, *n* = 8) or AMPA and NMDA-receptor mediated (0.5 μM DNQX, 5 μM AP5, *n* = 6) synaptic transmission were slightly reduced. Neuronal activity was stable throughout the recording for each condition (*Figure 7—figure supplement 1A,B*). At the single neuron level, AP firing was irregular, in line with our in vivo results (*Figure 7E,F*; *Figure 7—figure supplement 1C,D*). Temporal clustering was present under normal conditions (ACSF) but was reduced during disinhibition or disfacilitation (*Figure 7G*, PTX and DNQX/AP5, respectively). An intermediate level in correlated AP firing was found under normal conditions (*Figure 7H*). Correlations between neighboring and distant neurons were highly similar and as expected increased during disinhibition but decreased during disfacilitation (*Figure 7H*). As described in our in vivo results, the mean rate smoothly declined with increase in $λ_{thr}$ (*Figure 7—figure supplement 1E*), and the number of AP cascades of PN groups peaked in rate at an intermediate threshold $λ_{thr}$ (*Figure 7—figure supplement 1F*) for all three conditions. When processed at the corresponding $λ_{thr}^{max}$, cascade sizes under normal conditions distributed according to a power law that was robust to changes in $λ_{thr}$ (*Figure 7I*, *left*). As expected, the power law was destroyed when spiking activity was shuffled (*Figure 7I*, *right*). As previously shown for LFP-based analysis (*Plenz, 2012*), AP-based cluster size distributions became strongly bimodal during pharmacological disinhibition and slightly bimodal during disfacilitation (*Figure 7J*, PTX and DNQX/AP5, respectively; *Figure 7—figure supplement 2*). YC2.60, while being sensitive to single APs, tends to saturate for very strong spike bursts (*Yamada et al., 2011*). In contrast, the GECI GCaMP3

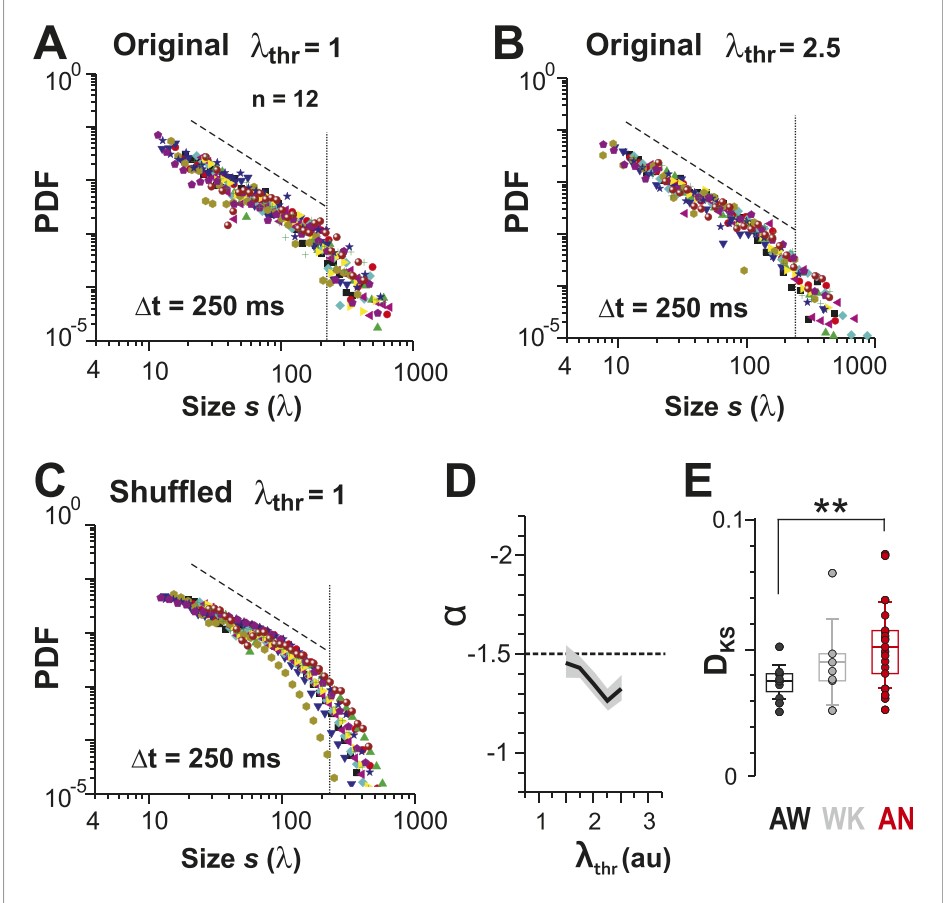

**Figure 6**. Identifying avalanche dynamics, that is, power law in clustering, using thresholding of the population rate vector (**Poil et al., 2012**). (**A**, **B**) Cluster size distributions for individual recordings (n = 12) following thresholding of the population rate vector at $\lambda_{thr}$ = 1 (**A**) and 2.5 (**B**). *Dashed line*: slope = −1.5. (**C**) Rate-preserved shuffling of λ in individual ROIs prior to calculation of population rate vector destroys the power law. (**D**) Power-law exponent, α, is relatively threshold-invariant. (**E**) Deviation from power-law dynamics at the population rate vector level increases with the transition from AW to AN (**\*\*p < 0.01).
The following figure supplement is available for figure 6:

**Figure supplement 1**. Scaling relationship between lifetime and size of spontaneous AP clusters supports neuronal avalanche dynamics (**Sethna et al., 2001**; **Friedman et al., 2012**).

(**Tian et al., 2009**) naturally has a higher threshold for AP detection (>3 APs) but reports even strong bursts linearly (**Yamada et al., 2011**) (**Figure 7—figure supplement 3A,B**). In line with our expectation of threshold invariance for LFP-based avalanches in the AW monkey (**Petermann et al., 2009**) and our YC2.60 measurements, we found that AP bursts measured with GCaMP3 were irregular at the single neuron level (**Figure 7—figure supplement 3C–H**), while AP cascades formed a clear power law (**Figure 7—figure supplement 3I,J**; n = 9 cultures). These in vitro results demonstrate neuronal avalanches to describe the spatiotemporal spike activity in L2/3 PN groups, that is, sensitive to the balance of excitation and inhibition and can be detected using high-threshold GECIs.

## Discussion

Here, we show that AP output from groups of PNs in superficial layers of cortex, while highly irregular at the single neuron level, assembles into scale-invariant neuronal avalanches at the local group level. In vivo, the emergence of avalanche organization in local pyramidal groups is linked to the AW state and is abolished under anesthesia. In vitro, this emergence occurs naturally and yet is sensitive to the

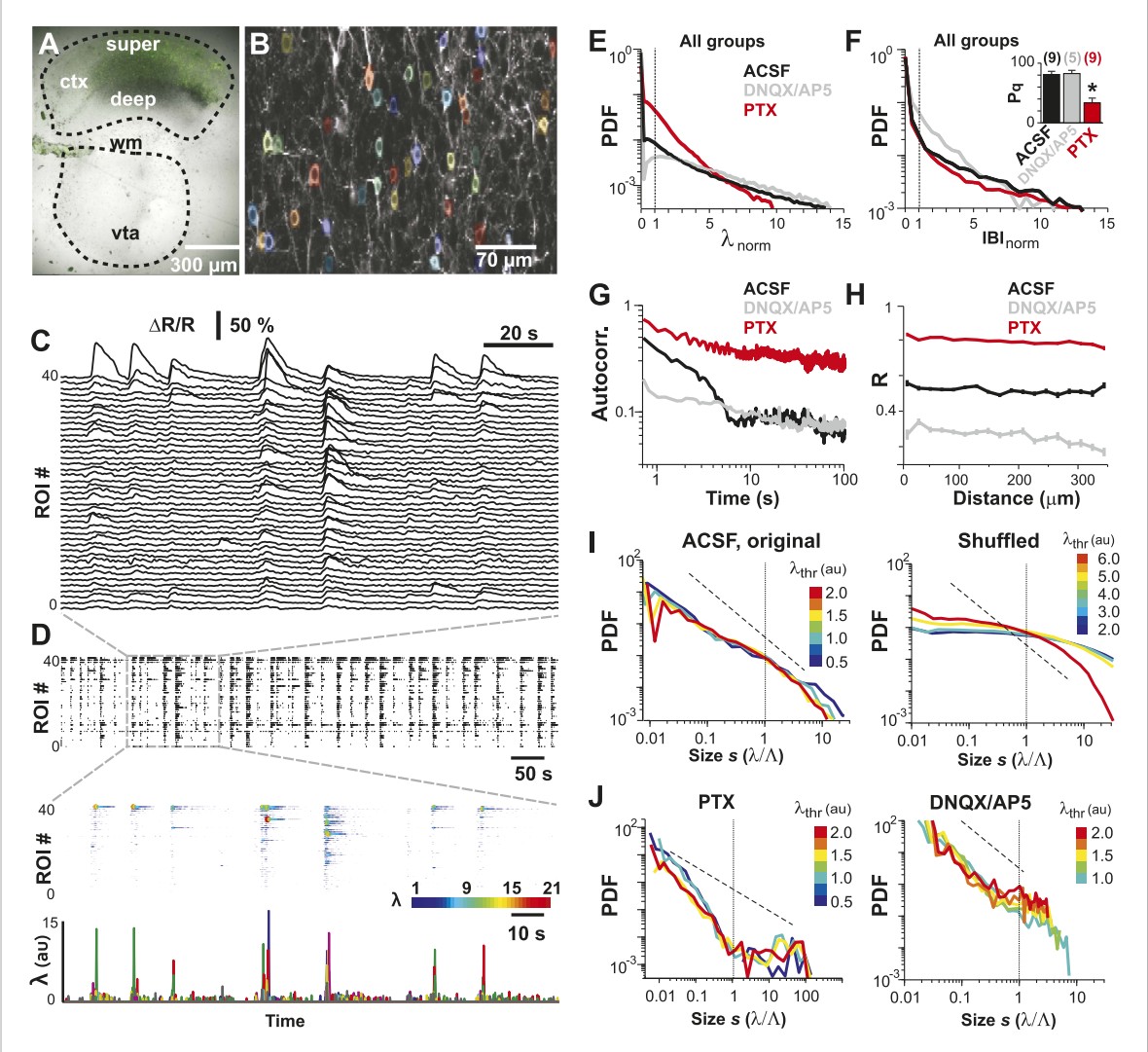

**Figure 7.** Spatiotemporal clustering in ongoing spiking activity recorded from groups of L2/3 PNs in vitro. (**A**) Organotypic cortex (*ctx*)-ventral tegmental area (*vta*) co-culture. YC2.60-expression in PNs from superficial (*super*) but not deep (*deep*) cortical layers. *wm*: white matter border location. (**B**) Single imaging plane containing a group of PNs with significant changes in *R* over time (colored ROIs). (**C**) Time course of ΔR/R for individual ROIs. (**D**) *Top*: Binary raster display of instantaneous spike rate estimate λ ($λ_{thr}$ = 0.1). *Middle*: Expanded period showing λ amplitudes. *Bottom*: Overplot of λ time course for individual (color coded) ROIs. (**E**) Distributions of $λ_{norm}$ for different pharmacological conditions (ACSF, DNQX/AP5, PTX). Dotted line, $λ_{norm}$ = 1. (**F**) Distributions of $IBI_{norm}$. Dotted line, $IBI_{norm}$ = 1. *Inset*: $P_q$ (number of recordings is indicated; *$p < 0.05$). (**G**) Mean λ autocorrelation function for individual PNs and different conditions. (**H**) Distance dependence of pairwise crosscorrelation in λ and different conditions. (**I**) *Left*: Power-law distributions in *s* for different $λ_{thr}$ (color scale) for normal condition (ACSF). *Right*: λ shuffling destroys power-law organization for normal condition. (**J**) Probability distributions in *s* for different $λ_{thr}$ (color scale) under PTX (*left*) and DNQX/AP5 (*right*). *Dashed lines* in **I** and **J**: slope = −1.5.

The following figure supplements are available for figure 7:

**Figure supplement 1.** (**A**, **B**) Mean λ rate and pairwise crosscorrelation are stationary over entire imaging session.

**Figure supplement 2.** Cluster size distributions under normal condition, disfacilitation (DNQX/AP5), and disinhibition (PTX) for YC2.60.

**Figure supplement 3.** Neuronal avalanche dynamics recorded in vitro using GCaMP3 in organotypic cortex cultures.

balance of excitation and inhibition. To date, avalanche dynamics have been demonstrated in numerous population signals, such as the LFP, EEG, MEG, and BOLD fMRI. These population signals, however, are ambiguous as to their site of origin, their emergence from non-neuronal or neuronal

elements, and their sensitivity to subthreshold and suprathreshold neuronal activities. Such ambiguities put into question whether computational benefits attributed to avalanche dynamics, or critical state dynamics in general (*Shew and Plenz, 2012*), extend to the local cortical microcircuit with computations taking place between PNs and local interneurons. Our results provide the first essential step in answering these functional questions at the local microcircuit level by demonstrating that, in the awake animal, local groups of PNs in layer 2/3 spontaneously exhibit highly variable AP clusters that organize as neuronal avalanches. By restricting our analysis to PNs, we avoid inflation of spike cluster from strongly firing interneurons and focus on the local output of a cortical microcircuit. The sensitivity of AP avalanches to anesthesia and the E/I balance demonstrates that PNs as well as interneurons are required to establish these dynamics. In light of the methodological and conceptual complexity of the current work, in what follows, we will first discuss methodological and technical aspects of our experimental design and avalanche analysis followed by a discussion of the concepts of criticality in the context of cortical AP activity at the microcircuit level.

## Dynamic range and linear estimation of instantaneous firing rates using the GECI YC2.60

GECIs derived from YC are fluorescence resonance energy transfer sensors, that is, the binding of calcium ions decreases fluorescence for short wavelengths while it increases fluorescence in the longer wavelength range (*Nagai et al., 2004*). The anti-correlated change at two wavelengths, captured in the ratiometric signal, increases the signal-to-noise ratio and naturally reduces global signal artifacts, such as those caused by movement of the animal. Our cell-attached recordings also demonstrate linear reporting of AP bursts, measured for up to 28 Hz (i.e., 7 APs/250 ms; *Figure 1—figure supplement 1B*). This observation suggests that the estimation of AP burst strength in our recordings is only weakly effected by saturation (*Lütcke et al., 2010*). The lack of impact of any potential saturation effect for YC2.60 on our main results was also demonstrated by our GCaMP3 in vitro recordings. GCaMP3 is relatively insensitive to small AP bursts and does not show saturation for relatively strong bursts, which introduces a natural $\lambda_{thr}$ of ~3 APs in our data acquisition (*Figure 7—figure supplement 3A*). To summarize, our choice of YC2.60 provided us with a GECI that exhibited an excellent dynamic range, that is, high signal-to-noise ratio, reasonable temporal resolution, high probability of single AP detection, and linear mapping of instantaneous firing rates up to ~20 Hz.

## Thresholding of continuous firing rate estimates and estimation of avalanche measures at various temporal resolutions

The identification of neuronal avalanche dynamics has been found to be robust to a wide range of thresholds for local event detection. As demonstrated in awake non-human primates, the amplitude of the negative LFP (nLFP), which identifies local synchronization, monotonically increases with the number of extracellular spikes recorded at the same microelectrode (*Petermann et al., 2009*). Thus, when identifying nLFPs on the microelectrode array using a threshold, a high threshold includes activity from local neurons that fire strongly, whereas a low threshold also includes weak local neuronal activity. It was found that for LFP-based avalanches, the threshold to identify nLFPs can vary over many standard deviations of the ongoing LFP fluctuations, yet the essential power law in avalanche sizes is maintained, despite a large decline in the nLFP and cascade rates with increase in threshold (*Petermann et al., 2009*). A similar threshold robustness was found for human avalanche dynamics based on the MEG (*Shriki et al., 2013*) or BOLD signal in the fMRI signal (*Tagliazucchi et al., 2012*). This is in line with simulations demonstrating that thresholding does not affect avalanche size distributions (*Lasse et al., 2009*). Threshold independence allows large local events (which are less frequent than small events) to be properly identified for avalanche analysis even at low temporal resolutions. In the current study, by increasing the local firing threshold, we isolated strong, local events in time and space, thus allowing for concatenation of significant local events into avalanches. At higher temporal resolution, the firing threshold can be lower as even small local firing can be properly separated in time (*cf. Figure 3B*). As was shown in the original publication on avalanches, at a fixed spatial resolution, prolonging/shortening Δt increases/decreases concatenation leading to a systematic change in the power-law slope of avalanches' sizes while maintaining the power-law signature (*Beggs and Plenz,*

*2003*). Similarly, the distribution of ROIs within an imaging frame establishes a spatial sampling matrix and correspondingly, the power-law slope was found to change accordingly with change in the frame rate (cf. *Figure 3E*).

## Spatial undersampling of cortical neuronal populations and the effect on avalanche parameters

Both our in vitro and in vivo recordings demonstrate avalanche dynamics to capture the organization of spontaneous spiking in a coarsely sampled sub-network of L2/3 PNs. The plane of imaging captured about 25–35 labeled PNs within a cortical area of $0.15 \pm 0.05$ mm$^2$. At a neuronal density of $\sim 100,000$/mm$^3$ for L2/3 in the adult rat (*Meyer et al., 2010*), we therefore expect $\sim 300$ PNs within a focal plane. Thus, we are recording from $\sim 10\%$ of PNs within our field of view. Simulations of spike avalanches using a branching process (*Ribeiro et al., 2010*, *2014*) or models of self-organized criticality (*Priesemann et al., 2009*) indicate that coarse spatial subsampling affects proper sampling of avalanche size distributions. Previous attempts to identify spike avalanches in the awake monkey found log-normal distributions, despite robust power laws in nLFP-based avalanches (*Petermann et al., 2009*). Such log-normal distributions have been suggested to indicate slightly subcritical dynamics in spike avalanches (*Priesemann et al., 2014*). However, our results demonstrate a power law in spike avalanches in the awake, but not anesthetized state, within our imaging frame. Because both states should be affected similarly from subsampling, we conclude that subsampling within our cortical field of view is not a major factor in our recordings to identify the power law in avalanche sizes. This could be because, in the above studies, the sampling density of spikes achieved by microelectrode arrays was orders of magnitudes lower than in the current study. On the other hand, because our cranial window only captured a small region of the cortex, we expect to see a cut-off in the avalanche size distribution, as reported in numerous experimental findings on avalanches based on the nLFP (*Beggs and Plenz, 2003*; *Petermann et al., 2009*; *Yu et al., 2014*) or MEG (*Shriki et al., 2013*) and simulations (*Priesemann et al., 2009*; *Ribeiro et al., 2010*; *Priesemann et al., 2013*; *Ribeiro et al., 2014*; *Yu et al., 2014*). Here, we show for the first time that this cut-off also holds for AP avalanches.

## Irregular firing with CV > 1, 'noise', and neuronal avalanche dynamics

Traditionally, irregular AP firing in individual PNs has been viewed as evidence that cortical dynamics are noisy and thus require averaging in time or across neurons to separate signal from noise (*Softky and Koch, 1993*; *Shadlen and Newsome, 1998*; *London et al., 2010*). Our results confirm that indeed single neuron firing is irregular in both the anesthetized and AW state. We found a lower firing rate, less irregularity, and less temporal clustering under anesthesia compared to the AW state. Yet, these differences did not specify any particular organization of the AW state, in particular, given that the average pairwise correlation between PNs did not differ between the two states. This latter finding suggested at first glance no difference in the average neuronal interactions. Only when temporally contingent spatial spike clusters in the neuronal population were taken into account, did the specific scale-invariant avalanche organization of the AW state become evident. Thus, irregularity of single neuron firing by itself does neither preclude nor determine avalanche dynamics.

Several neuronal network simulations have demonstrated the coexistence of avalanche dynamics and highly irregular firing at the single neuron level. Using a stochastic model of spiking neurons, Benayoun et al. show the coexistence of irregularity and avalanche dynamics close to a critical transition (*Benayoun et al., 2010*). Recent deterministic network models that incorporate synaptic plasticity also demonstrate how such irregularity robustly co-exists with avalanche dynamics, even in the absence of stochasticity (*Stepp et al., 2014*), yet demonstratively at a phase transition, where the addition of a single spike led to exponentially deviating network trajectories. Several computational studies in non-leaky as well as leaky integrate and fire networks have demonstrated the emergence of spike avalanches at or near a phase transition, where irregular spikes form clusters whose size distributions follow a power law (*Chen et al., 1995*; *Eurich et al., 2002*; *Levina et al., 2007*, *2009*; *Millman et al., 2010*). Thus, while irregular firing might be ubiquitous in many systems and during different brain states, it is in the AW state where irregularity combines with avalanche dynamics suggestive of a system residing near a phase transition. Our work, by providing detailed parameters on firing rate distributions, event count, interburst statistics, and correlations, should provide new experimental guidance to inform more realistic neuronal network models on critical spiking dynamics.

Computational models of cortical networks typically derive irregular spiking by establishing a balance in fast E/I (*Shadlen and Newsome, 1998*; *Renart et al., 2010*), which allows inhibitory and excitatory currents to track each other closely in time, resulting in an active de-correlation in spiking (*Renart et al., 2010*). However, since independent, stationary Poisson processes are insufficient to explain the high variability of spiking (CV > 1) observed typically in vivo (*Softky and Koch, 1993*; *Shadlen and Newsome, 1998*), alternative mechanisms beyond external Poisson inputs (*Brunel, 2000*) have been proposed to further increase variability, such as intrinsic chaotic dynamics (*Van Vreeswijk and Sompolinsky, 1996*; *Sussillo and Abbott, 2009*; *Ostojic, 2014*), conductance-based synapses (*Kumar et al., 2008*), clustered network architecture (*Litwin-Kumar and Doiron, 2012*), external synchronous inputs (*Stevens and Zador, 1998*), and 'doubly stochastic' approaches using non-stationary Poisson processes (*Churchland et al., 2010*), among others. Our results confirm the high variability of single neuron firing in the AW state with CV > 1. We suggest that the significant increase in CV beyond 1 for the AW state (*cf. Figure 2C*) is in line with the general notion that critical systems operate in a fluctuation-dominated regime, that is, high irregularity encountered at the single neuron level in the AW state might arise from large fluctuations that naturally occur when system dynamics approach a critical point (*Chialvo, 2010*; *Fraiman and Chialvo, 2012*; *Meisel et al., 2015*).

## The AW brain state and avalanche dynamics

Most in vivo studies reporting avalanche dynamics have been conducted in the AW animal for example, non-human primates (*Petermann et al., 2009*; *Yu et al., 2011*) or AW human subjects (*Fraiman and Chialvo, 2012*; *Palva et al., 2013*; *Shriki et al., 2013*). However, it was not clear whether avalanches also arise under anesthesia. In fact, avalanche analysis in deeply anesthetized cats or rodents reveals typical deviations, such as unusually shallow power laws based on the LFP and log-normal distributions of extracellular unit clusters (*Hahn et al., 2007*; *Ribeiro et al., 2010*). In these latter studies, however, the effect of anesthesia was difficult to separate from technical aspects, such as subsampling, which can affect spike clusters. In the present study, we (1) increased the number of neurons typically recorded with microelectrodes within an area of 200 × 200 μm by 1–2 orders of magnitude using 2-PI, (2) extracted spike clusters from a well-defined population of PNs, (3) used an exceptionally sensitive GECI to identify even single spikes, and (4) transitioned the animal between the anesthetized and the AW state. This approach provided us with the necessary precision and sensitivity to demonstrate the increasing deviation from a power law in size distribution that occurs even under light anesthesia (*cf. Figures 5B, 6E*). Our results suggest that avalanche dynamics might provide a precise 'fingerprint' to delineate the transition from the anesthetized state to the fully AW state. This delineation might be helpful in further quantifying different aspects of the AW state. For example, when studying avalanche dynamics in normal subjects, the degree of sleep deprivation was found to correlate positively with deviations from avalanche dynamics (*Meisel et al., 2013*). Intracranial recordings in human subjects have also shown small changes in 'vigilance' with changes in avalanche dynamics (*Priesemann et al., 2013*).

## Critical dynamics and near-critical dynamics

Avalanche dynamics were originally defined by (1) cluster sizes in LFP activity in vitro that follow a power law with slope close to −1.5, (2) a power law in life time distribution with slope of −2, and (3) a critical branching parameter of 1 (*Beggs and Plenz, 2003*). All three aspects have been demonstrated in spike clusters from layer 2/3 PNs in the present study. Detailed correspondences with avalanche work include the cut-off in size distributions beyond system size, originally identified in the LFP (*Klaus et al., 2011*), and clearly visible in the steep drop in spike cluster distributions beyond system size in the present study (*cf. Figures 3C, 4*). We obtained a branching parameter slightly smaller than 1 at the power law slope of −1.5, which might be due to the small neuronal group size recorded from. We also confirmed recent demonstrations of scale-invariance based on the collapse of avalanche waveforms (*Sethna et al., 2001*; *Papanikolaou et al., 2011*; *Friedman et al., 2012*). These measures combined strongly suggest that AP output of PN groups reflects critical dynamics in the AW state.

Avalanche measures have to be carefully evaluated for potential pitfalls. For example, power laws that arise from non-critical dynamics have been reported (*Touboul and Destexhe, 2010*), yet those power laws exhibit slopes of −10 to −50. In contrast, the slope of power laws for avalanche dynamics is typically more shallow than −2, that is, these distributions do not have a mean and display

unbounded variance, that is, non-existing first and second moments. Similarly, the upper cut-off in avalanche size distributions has sometimes been included into statistical fits. However, this upper cut-off arises from finite-size effects and needs to be disregarded for fitting (*Yu et al., 2014*), otherwise, statistical tests (*Langlois et al., 2014*) can be misdirected to fit the cut-off only (*Clauset et al., 2009*; *Dehghani et al., 2012*).

While our results clearly demonstrate that cortical dynamics approach a scale-invariant, that is, power-law organization in the AW state, the precise distance to the critical point is not known. In simulations, complex–hierarchical modular architectures of cortical networks have been shown to support critical dynamics (*Wang et al., 2011*). Such architectures, however, when combined with near-critical dynamics, can 'trap' activity (*Rubinov et al., 2011*; *Friedman and Landsberg, 2013*) inducing heavy-tail size distributions approximating power laws, or in general, extend the region in which critical-like behavior is observed by establishing so-called Griffiths phases (*Moretti and Muñoz, 2013*). Whether the layer 2/3 network can exhibit 'true' critical dynamics has also been called into question on the grounds that this would require a system to be placed exactly at the critical point, which is only possible for fine-tuned, 'conservative' systems (*Juan et al., 2010*). Cortical networks, while being critical in the long-term, could show subcritical transients. Subcritical dynamics have been invoked to explain findings from intracranial recordings in humans (*Priesemann et al., 2013*). In network simulations, a transition from subcritical to critical dynamics has been shown to benefit information processing (*Tomen et al., 2014*). Our results demonstrate that in the relatively fast transition from the anesthetized to the AW state, cortical dynamics more closely approaches or enters a critical regime. Finally, a dimension not employed in the current work is the specific temporal correlation structure of avalanches, which reveals scale-invariance and differs significantly from disinhibited or disfacilitated networks (*Lombardi et al., 2012*, *2014*; *Plenz, 2012*).

## In vitro results experimentally establish threshold-invariant avalanche dynamics in L2/3 PNs regulated by the E/I balance

Our in vitro results for the first time demonstrate that avalanche dynamics also describe the organization of AP patterns in a well identified neuronal population, local groups of L2/3 PNs in isolated cortex preparations. This overcomes previous limitations of in vitro studies in which cell identities and signal composition were largely unknown. The dependency of L2/3 avalanches on a $GABA_A$ antagonists and glutamate antagonists supports theories that indeed the E/I balance is important to establish avalanche dynamics in cortical networks. Our findings further demonstrate that in isolated cortex, avalanche dynamics is the natural organization that describes PN spiking in the absence of any inputs. Our in vitro recordings resulted in the collection of spike activity over longer periods in time compared to in vivo. This allowed us to demonstrate that GECIs, such as GCaMP3, with a natural threshold for spike burst imaging and which are less prone to saturation can also be used for avalanche detection (*Figure 7—figure supplement 3*), a direct experimental confirmation of simulations that showed thresholding does not affect avalanche size distributions (*Lasse et al., 2009*). Our in vitro results also provide additional benchmarks to which to compare in vivo avalanche dynamics. For example, pairwise correlations in vitro for avalanche dynamics are significantly higher compared to in vivo. While these values take an intermediate position compared to those found for the disinhibited or disfacilitated state (cf. *Figure 7H*), it is clear that absolute pairwise correlation values do not predict avalanche dynamics. Our findings transcend previous reports on the spontaneous formation of stable activity patterns in isolated cortical networks, such as the acute slice (*Sanchez-Vives and McCormick, 2000*; *Beggs and Plenz, 2003*; *Cossart et al., 2003*) and slice culture (*Beggs and Plenz, 2003*) or in vivo (*Miller et al., 2014*), which suggest the presence of 'attractor' states (*Beggs and Plenz, 2004*; *Ikegaya et al., 2004*; *Miller et al., 2014*). In fact, our work shows that AP patterns of L2/3 PNs form a specific subset of patterns in which sizes relate to each other in a scale-invariant manner.

## Avalanche spike output and potential underlying subthreshold activity

It is expected that a strongly fluctuating yet specific spatiotemporal organization in spikes will translate into correspondingly precise inputs in nearby PNs. Fast, subthreshold fluctuations in the intracellular membrane potential have been shown to translate into precisely timed action potentials (*Bryant and Segundo, 1976*; *Mainen and Sejnowski, 1995*) and that such spike precision carries information about the input (*Cecchi et al., 2000*). It is, therefore, reasonable to expect that spike avalanches translate into

fast voltage-fluctuations, which in turn generate precise spike outputs, thereby maintaining avalanche dynamics in the cortical microcircuit. Recent findings using voltage-sensitive dyes in layer 2/3 PNs indeed demonstrate neuronal avalanche dynamics to emerge in the AW mouse (*Scott et al., 2014*). Signals from voltage-sensitive dyes are proportional to the surface-to-volume ratio of the cellular compartment in which the dye is localized, and thus they preferentially report subthreshold intracellular membrane potential fluctuations (*Plenz and Aertsen, 1993*). This is in contrast to intracellular calcium reporters, which mainly report suprathreshold activity. Our study thus complements reports of neuronal avalanche dynamics in the input to PNs by demonstrating neuronal avalanches in the spike output of PNs in superficial layers in the AW state.

The emergence of neuronal avalanches at the neuronal group level does not exclude a role for critical dynamics at the cellular and subcellular level. The intermittent bursting behavior of single isolated neurons in response to stimulation (*Gal et al., 2010*; *Marom and Wallach, 2011*; *Gal and Marom, 2013*) suggests critical dynamics in the form of a low dimensional phase transition to control spike generation. Recent experimental demonstrations of critical slowing down as the membrane potential approaches spike threshold demonstrate critical dynamics to profoundly affect the AP generation (*Meisel et al., 2015*). Similarly, long-term fluctuations and power-law relationships have been reported for sodium channel gating (*Toib et al., 1998*). Thus, while spike avalanche dynamics emerge at the neuronal group level, the underlying mechanisms are likely to involve specific dynamical properties at the single cell and subcellular level.

## Final conclusions

The identification of avalanches in the main excitatory cell type that constitutes the mammalian cortex establishes the strongest proof to date that avalanche dynamics provide the guiding principles for propagation of cortical activity. This finding should have a number of consequences for a cellular understanding of cortical network activity. First, optimization principles in information processing identified for avalanche dynamics at the population level of cortex should be directly applicable to the interaction of PNs (*Beggs and Plenz, 2003*; *Bertschinger and Natschlager, 2004*; *Kinouchi and Copelli, 2006*; *Rämö et al., 2007*; *Nykter et al., 2008*; *Shew et al., 2009*; *de Arcangelis and Herrmann, 2010*; *Shew et al., 2011*). Specifically, local layer 2/3 networks should exhibit maximal dynamic range to process layer 4 inputs and maximize mutual information between patterns elicited in layer 4 and superficial layers. Second, the nature of spontaneous, irregular firing in PNs profoundly influences theories on cortical function. For example, when these fluctuations in firing are considered to reflect noise, spatiotemporal averaging over neuronal populations can be used to enhance response encoding at the expense of temporal precision and neuronal identity (*Shadlen and Newsome, 1998*; *London et al., 2010*; *Renart et al., 2010*). On the other hand, we demonstrated that fluctuations in single neuron firing, amount to a scale-invariant order in active neuronal groups, suggestive of critical dynamics guiding single neuron firing. Accordingly fluctuations that arise from long-range spatiotemporal correlations between neurons should not be averaged (*Fraiman and Chialvo, 2012*) but instead need to be taken into account, for example, for theories on cortical population coding (*Averbeck et al., 2006*). Finally, it is well known that resting or ongoing activity profoundly influences stimulus responses (*Arieli et al., 1996*; *Sato et al., 2007*; *Luczak et al., 2009*; *Womelsdorf et al., 2012*). For example, evoked visual responses correlate strongly with ongoing activity shortly preceding the stimulus (*Arieli et al., 1996*). Long-term temporal correlations have been soundly established at the population level, such as the EEG (*Linkenkaer-Hansen et al., 2001*) and ECoG (*He et al., 2008*). Our findings suggest that the occurrence of a single 'spontaneous' spike or spike burst in the AW state correlates with activity in other PNs over time and distance in cortex, as quantified by the scale-invariant correlation structure established by neuronal avalanches. We suggest that this will be of particular importance in the context of 'noise correlations', which capture the non-stimulus induced correlation structure and tend to affect the decoding capability of a neuronal population (*Averbeck et al., 2006*; *Insabato et al., 2014*).

The emergence of scale-invariant order from the interaction of local elements is a hallmark of systems at criticality (*Plenz and Thiagarajan, 2007*; *Chialvo, 2010*; *Plenz and Niebur, 2014*). By demonstrating such scale-invariance to exist at the neuron level, we suggest that neuronal avalanches provide a unifying framework of cell assembly formation in cortex that ranges from local groups of neurons to the global scale of the brain.

## Materials and methods

All animal procedures were approved by the National Institute of Mental Health Animal Care and Use Committee and were carried out in accordance with National Institutes of Health guidelines.

### In utero electroporation

Timed-pregnant rats (Sprague Dawley, embryonic day 15.5 ± 0.5, Taconic Farms) underwent a laparotomy (1.5–4 % isoflurane anesthesia) during which 5–8 µg of purified plasmid DNA (Endofree Maxiprep kit, Qiagen, Germantown, MD), consisting of Yellow Cameleon 2.60 (YC2.60) (Mikoshiba Lab, RIKEN, Japan; [*Yamada et al., 2011*]) or GCaMP3 (*Tian et al., 2009*) subcloned into a pCAG backbone, was pressure-injected through the uterine wall into the frontal ventricle of one hemisphere using a fine point glass capillary. DNA was electroporated into cells of the subventricular zone (*Saito and Nakatsuji, 2001*; *Saito, 2006*) using platinum tweezertrodes (5 mm diameter; 5 square pulses, 45 V amplitude, 50 ms duration; ECM-830, Harvard Apparatus, Holliston, MA), predominantly labeling PNs in superficial cortical layers 2 and 3 (L2/3; see [*Saito, 2006*]; *Figure 1A*).

### Organotypic co-cultures

In utero electroporated pups (postnatal day [P] 1–3) were checked for expression of YC2.60 in dorsolateral cortex and used for the preparation of organotypic co-cultures, consisting of cortex and ventral tegmental area (VTA), as described previously (*Gireesh and Plenz, 2008*). In brief, coronal sections of cortex and midbrain were cut using a vibratome (VT100 S, Leica) under sterile conditions at 350 µm and 500 µm, respectively. Regions of cortex (up to 2 mm wide) containing all layers, and midbrain tissue containing the VTA, were excised and attached adjacent to each other on a glass coverslip. Co-cultures were grown under sterile conditions in standard culture medium in a roller tube arrangement and were used for electrophysiology and 2-photon imaging (2-PI) after 14–20 days in vitro (DIV).

### Head bar implantation, habituation, and craniotomy

For 2-PI in the AW animal, in utero electroporated rats were first identified by transcranial YC2.60 fluorescence observation at P1–3. Animals expressing YC2.60 were fitted with a custom-made, T-shaped stainless steel head bar at ~P21. To this end, animals were anesthetized (isoflurane: 4% induction, 1.5–2% maintenance) and mounted in a stereotaxic frame. After a midline incision was made in the scalp, the skull surface was cleared of membranes; adhesive luting cement (C&B Metabond, Parkell, Inc., Edgewood, NY) was applied contralaterally to the YC2.60-expressing hemisphere, followed by attachment of the head bar using Grip cement (Dentsply International Inc., Milford, DE). Rats were given an analgesic (Ketoprofen, 5 mg/kg s.c) for up to 2 days post surgery. Rats were habituated to the recording condition for up to 5 sessions post surgery. In each session, the rat was briefly anesthetized (<5 min of 2–3% isoflurane) and installed in the head fixation apparatus, which consisted of a plastic tube which loosely confined the rat's limbs without restricting breathing, a platform and a custom-made steel beam, which screwed into the head bar at one end and a fixed post at the other end, allowing horizontal, vertical, and axial freedom of movement to position the rat's head under the 2-PI objective. After awakening, rats were left in the apparatus for 10–20 min per session. Rats became comfortable with the recording condition after 3–5 habituation sessions. On the day of imaging, rats were subjected to craniotomy and cranial window implantation. Rats were anesthetized (isoflurane: 4% induction, 1.5–2% maintenance) and mounted in a stereotaxic frame. The location of the craniotomy was determined by observation of transcranial YC2.60 fluorescence and was usually found within sensorimotor and frontal cortex (from bregma: AP 0.5 ± 1.0 mm, ML 3.0 ± 0.5 mm). A section of the skull (~3–4 mm diameter) was removed using a dental drill and the underlying dura was resected. Care was taken not to damage any subdural blood vessels. The exposed brain was continuously irrigated with sterile saline. Finally, a glass coverslip was cut to the size of the craniotomy using a stylus, mounted on the opening using low melting point agarose (1–2% in sterile saline), and secured with Grip cement. Rats were given an analgesic (Ketoprofen, 5 mg/kg s.c) and allowed to recover for at least 3–6 hr before undergoing 2-PI.

### 2-Photon imaging

For in vivo 2-PI, rats (P27–35) that had undergone head bar implantation, habituation, and craniotomy were anesthetized (isoflurane: 4% induction, 1–2% maintenance), head-fixed, and

placed under a 2-photon microscope (25× objective, 1.05 NA, 1000 MPE, Olympus, Center Valley, PA). YC2.60 was excited at 840 nm (Chameleon Vision II, Coherent, Santa Clara, CA), and cpVenus and ECFP fluorescent emission were collected using 460–500 nm and 520–560 nm bandpass filters, respectively, separated by a 505 nm dichroic mirror. Once an imaging field containing up to ~40 neurons was located, 5–15 min long movies were recorded at a temporal resolution, Δt, of ~250 ms. Higher temporal resolutions (167 and 88 ms) were achieved by 2× and 4× line skipping, respectively (Olympus Fluoview software). To obtain movies in the waking (WK) and AW animal, isoflurane was turned off. Movies recorded >5 min after turning off isoflurane were classified as WK, and subsequent movies (>20 min after turning off isoflurane) were classified as AW. During WK and AW conditions, the behavioral state of the animal was monitored with an infrared (IR) camera (c525, IR filter removed, Logitech, Newark, CA). Periods of animal movement (which were minimized by habituation) generated an easily identifiable artifact in the ΔR/R calcium signal (see below) and were manually removed before analysis.

For in vitro 2-PI, cultures were submerged in oxygenated artificial cerebrospinal fluid (ACSF, bubbled with 95% $O_2$ and 5% $CO_2$) containing (in mM) 124 NaCl, 3.5 KCl, 10 D-glucose, 26.2 NaHCO$_3$, 0.3 NaH$_2$PO$_4$, 1 MgSO$_4$, and 2 CaCl$_2$ warmed to 32°C at a flow rate of 1 ml/min. Intracellular calcium dynamics of 15–80 spontaneously active PNs were imaged continuously within a 250 µm by 50–100 µm wide region for 5–20 min with a temporal resolution Δt = ~250 ms.

## Choice of GECI

For our experiments, we chose YC2.60 over related GECIs, such as D3cpv and YC3.60, for the following reasons. In general, single AP detection in vivo is still below 100% for yellow chameleons and related GECIs (Lütcke et al., 2010; Margolis et al., 2012). We excluded D3cpv, which shows single AP sensitivity, due to its saturation for small AP bursts (Wallace et al., 2008). YC3.60 is an interesting alternative to YC2.60 due to its higher $K_D$ and shorter decay time constant (0.5 s in vivo at physiological temperature; 0.8 s in vitro at room temperature [Yamada et al., 2011]). While a short decay constant allows for higher temporal resolution in imaging, YC3.60 is about 50% less sensitive to single APs compared to YC2.60. Given the relatively low spontaneous AP rate for neurons in superficial cortical layers in vivo (for review see [Barth and Poulet, 2012]), we, therefore, opted for YC2.60 with its somewhat longer decay constant of ~1–2 s. Our observation of 4–5% ΔR/R for single APs using YC2.60 in layer 2/3 (L2/3) PNs in vitro at 32°C is within the range reported for D3cpv in vitro (8.3% at room temperature) and in vivo (3.5%) (Wallace et al., 2008). It is also in line for YC2.60 single AP detection at 33°C in the acute slice, which shows a ΔR/R of ~4–5% and a decay time constant of ~2 s for a 10 AP burst (Yamada et al., 2011). For YC3.60, sensitivity and decay time constant were shown to be similar in vivo (Lütcke et al., 2010) and in vitro when measured at physiological temperature (Yamada et al., 2011). The range in similarities for YC GECIs suggests that our YC2.60 in vitro characterization at physiological temperature similarly predicts its performance in vivo. This is further supported by the insensitivity of coefficient of variation (CV) in our in vivo data to the changes in $\lambda_{thr} < 1$ (cf. Figure 2—figure supplement 1), which is in line with the binary detection operation of single APs for that range (cf. Figure 1—figure supplement 1A).

## Calcium imaging analysis

YC2.60-expressing PNs were visually identified either by high-average somatic fluorescence or high-somatic fluorescence CV, which captures relatively quiescent neurons with intermittent, sparse spiking activity and whose average somatic fluorescence remained low. Boxcar regions of interest (ROIs) were manually drawn around the somatic region of labeled neurons for which the nucleus was clearly visible within the cross-sectional somatic area. The boxcar was aligned with the outer perimeter of the neuron, and all pixels within the boxcar were taken for analysis. For YC2.60, the ratio, R, of cpVenus fluorescence to ECFP fluorescence was calculated for each ROI and each frame. The ratio measurement requires neuronal signals to be anti-correlated in the two wavelength bands, which allows for easy identification of non-signal artifact, for example, from small animal movements. A continuous function of fluorescence was calculated as ΔR/R = (R$_{ROI}$ − R$_0$)/R$_0$, where R$_{ROI}$ denotes the average fluorescence ratio within the ROI. The baseline fluorescence ratio, R$_0$, was defined as the median of R. ΔR/R was low-pass filtered (Kampa et al., 2011) (3Δt; symmetrical Gaussian kernel), and the instantaneous firing rate estimate, λ, in arbitrary units was obtained for each ROI using fast,

non-negative deconvolution (*Vogelstein et al., 2010*) (OOPSI package, Matlab, MathWorks Inc., Natick, MA). The decay time constant of the non-negative deconvolution for YC2.60 was ~1–1.5 s, as estimated from simultaneous cell-attached recording and 2-PI (n = 8 neurons). This value was also obtained by minimizing the summed square error between $R$ and a reconstructed $R'$, made by convolving an exponential decaying kernel with $\lambda$ over the range of $\tau = \Delta t - 15 \times \Delta t$ for neuronal populations recorded in vivo and in vitro (*Figure 1—figure supplement 1C*) and is in line with previous reports (*Yamada et al., 2011*). The a priori rate estimate for spiking was set to 1 Hz based on a meta-study (*Barth and Poulet, 2012*), and lower rates were examined systematically in the form of thresholding on $\lambda$ ($\lambda > \lambda_{thr}$). The standard deviation parameter was estimated for each $\Delta R/R$ using calcium amplitudes less than 5% $\Delta R/R$, corresponding to the average amplitude for single spike detection of YC2.60. Deconvolution parameters were determined, first, from loose-patch recordings (*Figure 1—figure supplement 1A,B*) and second, from minimizing the residual error in reconstructed $\Delta R/R$ traces from $\lambda$ (*Figure 1—figure supplement 1C*). Both methods yielded an optimal deconvolution time constant of ~1.5 s, in line with previous reports (*Yamada et al., 2011*).

ROIs with $\lambda_{avg} < 0.016$ (approx. less than 1 AP/min) were removed from the analysis (*O'Connor et al., 2010*). The average instantaneous spike rate estimate of $\lambda$ for each ROI was calculated for a range of $\lambda_{thr} = 0.1$ to 8 and included zero estimates for time bins $\Delta t$ with $\lambda \leq \lambda_{thr}$. The average burst strength of $\lambda$ was the mean of all bins with $\lambda > \lambda_{thr}$. Interburst intervals (IBIs) were defined as consecutive bins of length $\Delta t$ for which $\lambda \leq \lambda_{thr}$. Pairwise crosscorrelations were calculated in Matlab (Mathworks Inc.) using the function *corrcoef*. All other analyses, unless stated otherwise, were performed in Matlab using custom routines.

## Relationship between spontaneous AP bursts and intracellular calcium transients

In order to identify the relationship between the number of spontaneous APs and $\Delta R/R$ (YC2.60) or $\Delta F/F$ (GCaMP3), loose-patch, voltage-clamp recordings were performed in GECI-expressing PNs in organotypic cortex-VTA cultures (*Gireesh and Plenz, 2008*) at 14 DIV or later. Cultures were submerged in regular ACSF (flow rate, 1 ml/min) at 32°C. After forming a loose-patch on a visually identified YC2.60 (n = 8) or GCaMP3 (n = 8) labeled PN in L2/3, spontaneous, extracellular AP currents were recorded simultaneously with intracellular calcium transients using 2-PI. For each imaging frame with duration $\Delta t = 250$ ms, the number of spontaneous APs was correlated with $\Delta R/R$ as well as the corresponding peak amplitude and integrated area of the firing rate estimate $\lambda$ (*Figure 1—figure supplement 1A,B*). This relationship was also true for $\Delta F/F$ (data not shown). To obtain comparable $\lambda$ and IBI distributions for neurons with different average rates, $\lambda_{avg}$, the normalized rate $\lambda_{norm} = \lambda/\lambda_{avg}$ and normalized $IBI_{norm} = IBI/IBI_{avg}$ were used.

## Analysis of calcium transients from surrogate data for different temporal resolutions

The relationship of $\lambda$ and the real spike count for different temporal resolutions was evaluated using spike trains from 26 extracellularly recorded single units in superficial layers of somatosensory cortex from adult rat during the AW resting state in a separate experiment. For these recordings, we used a Neuronexus array with 8 short shanks and 4 electrodes per shank separated by 200 μm. The array was lowered into the cortex under visual control until the last electrode entered layer 1, anchored using dental cement, and the cranial opening was closed for chronic in vivo recordings. The array configuration and array insertion targeted unit activity from within the first 600–800 μm of cortical depth, which largely covers superficial layers in the adult rat. Unit activity was sampled at 30 kHz and sorted offline (Offline sorter, Plexon, Dallas, TX). Simultaneous LFP recordings from the same electrodes demonstrated nLFP-based avalanche dynamics in the AW state, further supporting superficial layers as the main recording sites (see e.g., [*Stewart and Plenz, 2006*; *Gireesh and Plenz, 2008*]; data not shown).

To obtain surrogate calcium traces at different temporal resolutions, the following three steps were performed: (1) spike trains were convolved using an impulse function with instantaneous 5% peak amplitude and an exponential decay of 1.5 s, parameters obtained from 2-PI (*Figure 1—figure supplement 1*). (2) The resulting calcium traces were sampled at 100 Hz, and uniform noise was added. The noise level for $\Delta t = 250$ ms was set to ±8%. Similar results were found for Gaussian noise. (3) Calcium traces were down-sampled to a final temporal resolution of $\Delta t = 250$, 167, or 88 ms.

In order to simulate the lower signal-to-noise ratios for smaller Δt, which resulted from line skipping during in vivo imaging, we adjusted noise levels by a factor of $\sqrt{2}$ and $\sqrt{4}$ for Δt = 167 and 88 ms, respectively. From the resulting calcium traces (*Figure 1—figure supplement 2A*), λ was estimated using fast, non-negative spike deconvolution (*Vogelstein et al., 2010*). This analysis showed that the mapping of λ to number of APs/Δt is linear and has the same slope (*Figure 1—figure supplement 2B*; all $R^2 = 0.99$).

## Neuronal avalanche analysis

Spatiotemporal clusters of L2/3 PN activity were defined by spiking activity in at least one ROI above a given threshold ($\lambda > \lambda_{thr}$) within the same or consecutive time bins of duration Δt (*Beggs and Plenz, 2003*). By definition, a cluster is flanked by empty bins, that is, all ROI have $\lambda \geq \lambda_{thr}$ (*Figure 3A*). The value of Δt was determined by the temporal resolution of 2-PI. The value of $\lambda_{thr}$ was chosen such that the number of cascades was maximized for a given recording (*Poil et al., 2012*) (*Figure 3B*). The size, $s_\lambda$, of a cluster was defined as the sum of λ across all active neurons, i = 1,…,k within the cluster, that is, $s_\lambda = \sum_{i=1}^{k}\lambda^i$. We found that $s_\lambda$ was proportional to the number of active neurons in a cluster, $s_k = \sum_{i=1}^{k}1$, by a factor given by the average rate across all ROIs, $\lambda_{avg}^{pop}$ ($R^2 = 0.98$). Therefore, $s_k$ and $s_\lambda$ provide similar information about avalanche sizes, as reported previously for nLFP-based avalanches (*Beggs and Plenz, 2003*). To compare across experiments with different number of neurons, N, (i.e., ROIs) cluster sizes $s_\lambda$ were normalized by the predicted cluster size limit, $\Lambda = N\lambda_{avg}^{pop}$ (*Klaus et al., 2011*; *Yu et al., 2014*). For visualization, cluster size distributions were logarithmically binned (30 bins). Neuronal avalanches were defined by their distribution in cluster size that follows a power law with an exponent close to −1.5, up to the cluster size limit. The branching parameter σ, the average ratio of the number of descendants to ancestors within a cascade (*Beggs and Plenz, 2003*), was estimated using participation counts from the first (ancestor) and subsequent (descendent) Δt bins within each cascade. Lifetime, T, was defined as the number of active time bins during the avalanche multiplied by Δt. A population thresholding approach on the instantaneous integrated population vector on λ, $\lambda^{pop}$, has recently been introduced as an alternative method to obtain the distribution of cascade sizes (*Poil et al., 2012*). For a given $\lambda^{pop}$ based on a chosen $\lambda_{thr}$, population thresholds were placed to search for the maximum number of cascade sizes. By including only those Δt bins whose member events sum above a minimum threshold, this method avoids issues in cascade concatenation when periods of empty Δt bins may be rare due to insufficient temporal resolution or when monitoring a high number of units.

## Power-law parameter estimation and statistical analyses

If not stated otherwise, power-law exponents were estimated by minimizing the Kolmogorov–Smirnov distance, $D_{KS}$, between the cumulative distribution functions (CDFs) of the data, $C_{data}(s)$, and the power-law model, $C_\alpha(s)$ (*Klaus et al., 2011*):

$$\hat{\alpha} = arg \min_{\alpha} D_{KS},$$

and

$$D_{KS} = \max_s |C_{data}(s) - C_\alpha(s)|.$$

The power-law model for $s_{min} < s < s_{max}$ is given by $P_\alpha(s) = cs^\alpha$, where $c = (\alpha+1)/(s_{max}^{\alpha+1} - s_{min}^{\alpha+1})$ is the normalization constant. The corresponding CDF is then defined by $C_\alpha(s) = \int_{s_{min}}^{s} P_\alpha(s)ds$. $s_{min}$ was set to the smallest observed avalanche size. The upper bound, $s_{max}$, was set to the predicted cluster size limit $\Lambda = N\lambda_{avg}^{pop}$ at which cluster size distributions started to deviate from a power law (*Figure 3C*; see [*Yu et al., 2014*]). $D_{KS}$ was also used to compare the goodness of the power law fit across different conditions (*Figure 5B*; *Figure 6E*). For the comparison of the power law vs the exponential distribution, the log-likelihood ratio (LLR) was calculated and parameter estimates were obtained by likelihood maximization (*Clauset et al., 2009*; *Klaus et al., 2011*):

$$LLR(x) = l(\alpha|x) - l(\gamma|x),$$

where $l(\alpha|\boldsymbol{x}) = \sum_{i=1}^{n}\ln P_\alpha(x_i)$ is the log-likelihood of observing the sample vector, $\boldsymbol{x} = x_1,…,x_n$, assuming the power-law model $P_\alpha(s)$, and $l(\gamma|\boldsymbol{x}) = \sum_{i=1}^{n}\ln P_\gamma(x_i)$ is the log-likelihood of observing $\boldsymbol{x}$ assuming an exponential model $P_\gamma(s) = ce^{-\gamma s}$ with c being the corresponding normalization constant. The *LLR* obtains positive values if the data are better fit by a power law compared to an exponential

distribution, and negative values if the exponential distribution yields the better fit. The p-value for determining statistical significance is given by (*Clauset et al., 2009*; *Klaus et al., 2011*):

$$p = \text{erfc}\left(\frac{|LLR|}{\sqrt{2n\sigma^2}}\right),$$

where

$$\sigma^2 = \frac{1}{n}\sum_{i=1}^{n}\left[(l(\alpha|x_i) - l(\alpha|\mathbf{x})/n) - (l(\gamma|x_i) - l(\gamma|\mathbf{x})/n)\right]^2.$$

## Statistical analysis

One-way analysis of variance (ANOVA) was used for multiple comparisons with Bonferroni *post hoc* test if not stated otherwise. Error bars and shaded areas around averages denote standard error of the mean.

## Acknowledgements

We thank Samantha Chou for help with training of the rats, Dave Ide for help with the in vivo set up, Craig Stewart for help with preparing organotypic cultures, and Dr Rajarshi Roy and members of the Plenz lab for lively discussions. This research was supported by the Intramural Research Program of the National Institute of Mental Health, NIH, USA.

## Additional information

### Funding

| Funder | Grant reference | Author |
| --- | --- | --- |
| National Institute of Mental Health (NIMH) | Division of Intramural Research | Timothy Bellay, Andreas Klaus, Saurav Seshadri, Dietmar Plenz |

The funder had no role in study design, data collection and interpretation, or the decision to submit the work for publication.

### Author contributions

TB, AK, Conception and design, Acquisition of data, Analysis and interpretation of data, Drafting or revising the article; SS, Acquisition of data, Drafting or revising the article; DP, Conception and design, Analysis and interpretation of data, Drafting or revising the article

### Author ORCIDs

Andreas Klaus, http://orcid.org/0000-0002-4133-351X

### Ethics

Animal experimentation: All animal procedures were approved by the National Institute of Mental Health Animal Care and Use Committee (protocol # LSN-01) and were carried out in accordance with National Institutes of Health guidelines.

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
