## [Decision Letter]

Thank you for sending your work entitled “Avalanche dynamics emerge from irregular spiking of pyramidal neurons in the awake resting state” for consideration at *eLife*. Your article has been favorably evaluated by Timothy Behrens (Senior editor) and three reviewers, one of whom, Frances Skinner, is a member of our Board of Reviewing Editors

The Reviewing editor and the other reviewers discussed their comments before we reached this decision, and the Reviewing editor has assembled the following comments to help you prepare a revised submission.

Overall, all of the reviewers felt that the work was novel, of superb methodological quality and potentially influential (in linking avalanche dynamics to a global physiological state and demonstrating in vivo the relation with spikes). However, it was strongly felt that extensive rewriting and presentation is needed to allow the work to be appreciated and understood more widely. Here are some specific suggestions and other details are provided below from reviews.

1) Be clear about the uniqueness of the paper without overstating. That is, two aspects of avalanche dynamics are shown:

(i) A first demonstration of avalanche dynamics linked to a global physiological state (wake vs. anesthetized);

(ii) A first demonstration of relations between macroscopic avalanche dynamics and spikes, in-vivo. All previous reports relied on population activity measures, not spikes. (A couple of papers by the Chiappalone group, of Genova, did show relations of spiking activity to avalanche dynamics in-vitro, not in-vivo; but these dynamics, due to particular choice of binning in the analysis, were mostly related to activity within synchronized population bursts).

2) Provide a bit more background and context for the work. In other words, explain things (i.e., background theoretical assumptions/limitations etc.) better for the general reader. Specifically, while many citations are provided (e.g., consider the multitude in the first paragraph), there are not straightforward explanations of how neuronal avalanches, power law and criticality are interlinked, and what this actually means in terms of brain functioning. For example, the “being critical of criticality” Beggs and Timme Frontiers 2012 article does a really nice job of these concepts. Perhaps an additional subsection in the Introduction could be written.

3) Alternative interpretations should be mentioned and considered, and the potential functional implications of findings should be discussed, via additional paragraph(s) in the Discussion. This should include an explicit discussion on the meaning of the present results in the context of debates that concern whether-or-not actual information processing during tasks operates in a “critical” regime.

4) Overstatements should be removed. For example, in Abstract, reduce the tone where avalanche dynamics are attributed “optimal” for information processing. The results do not relate to “function”, but rather to an awake state. Assuming that other types of neurons (or pyr cells at other layers) have not been measured, do not over-state the relation of avalanche dynamics to pyramidal cells.

Reviewer #2:

1) The manuscript “Avalanche dynamics emerge from irregular spiking of pyramidal neurons in the awake resting state” by Bellay et al. demonstrates that the spatial and temporal correlation of (suprathreshold) firing events in superficial layer pyramidal cells can be described as avalanches in the awake state to a larger degree than during anaesthetized and awaking states. The authors first show that their estimate of instantaneous firing (λ) from calcium signals is linked to action potental firing (Figure 1) and they show that neurons fired irregular with long periods of quiescents in all states and showed largest variabilitiy (CV) during the awakening period (Figure 2). More importantly, the study reports that neural firing (indexed as λ) showed only low correlations with somewhat more correlated firing during awakening that during the awake state. Interestingly, avalanche dynamics was more pronounced during the awake state (Figure 5—figure supplement 1, with the so called avalanche life time distributions following a power law as described before). The authors also show in in vitro recordings that the temporal correlations are reduced (with inhibition) or enhanced (during disinhibition) but remained intact during moderate reductions of GABAa, AMPA or NMDA synaptic activity (Figure 7). Spatial clustering was absent in these in vitro recordings.

The manuscript convincingly demonstrates avalanche dynamics in superficial layer pyramidal cell firing that is state dependent with enhanced avalanche dynamics during the awake state. This insight follows from an extensive characterization for the temporal and spatial clustering of firing events from in-vivo calcium recordings. Methods, experimental approach are all state-of-the-art.

However, the manuscript did not attempt to quantify why or how the avalanche dynamics is influencing information processing in the circuits. As such the study provides more of a proof of principle, using sophisticated tools, that temporal and spatial clustering's are evident in calcium imaged firing rate dynamics of large populations of neurons (as it is evident in all population metrics studied so far to my knowledge).

2) If possible, the study would increase its appeal by showing how (in quantitative terms) e.g. a stimulus pulse, (or maybe a spontaneous bursting event) directly predicts some events/dynamics at later time points (or whether it predicts the awake state). Such an approach would help to identify whether there are actual information processing effects resulting from events that are characterized by avalanche dynamics.

3) There are some aspects of the manuscript that deserve consideration and which are listed here:

The Title needs some clarification. In the text authors mention that avalanches depend on balance of excitation and inhibition. Since inhibitory neurons are not identified in the current work but seem to play a role, it seems a foregone conclusion to state that it is the firing of pyramidal cells that lead to avalanches as is suggested by the Title. Do the authors imply that avalanche dynamics would not be evident when recording interneuron activation patterns?

As far as I see the authors infer that distributions follow scale free power law behavior simply from the fact that the fits are significant and that shuffling of the data do not result in significant fits. While this may be the standard in the field I am worried that there are no alternative hypotheses offered or tested. Shouldn't a statistical test compare two functions in their degree of predicting the distribution? I ask this because it is one thing to describe the dynamics of the system as satisfying a scale free power law, but the interpretation and inferences would be different if other non-scale free functions describe the data in similar good ways. What would happen to the criticality perspective if future studies would show that the dynamics would follow only “somewhat” a power law, but that the fine scale dynamics can be as well described by scale specific function? This is probably not only unclear to ‘non-technical’ readers.

The authors estimate the relation of the inferredλ-firing with actual firing rates measured with a Neuronexus array (in the subsection headed “Analysis of calcium transients from surrogate data from different temporal resolutions”). It is crucial to add information on how likely this array records neurons from similar layers as in the calcium measurements. Are firing rates from L2/3 different between layers? What is the firing rate in VTA neurons and their relation to λ. Some more explicit information on this would be helpful.

*Reviewer #3*:

The concept of neural avalanche dynamics was introduced by Beggs and Plenz over a decade ago, and continues to serve as a fruitful metaphor in analyses of experimental data and in development of theory. It is an appealing concept from both the mechanistic and functional aspects: providing clues for the origin of scale invariance in the statistics of neural network activity, as well as offering a point of view on the enrichment of network activity patterns by means of matched excitation-inhibition ratios.

In this paper Belly et al. provide convincing evidence for the occurrence of avalanche dynamics—scale invariant cascades of neuronal spiking activities—in the rat superficial cortical layers. Moreover, they show that these dynamics are affected by a global state of the animal—anesthetized vs. awake—and provide a nice link between single neuron irregular firing statistics and population statistics.

I have one main comment that concerns paper's structure. I do not understand why the authors include in this paper the in-vitro results. They seem irrelevant to the point made, and even the authors themselves do not mention these results in the Abstract, nor discuss these data in the Discussion section. The only mentioned justification for the choice (i.e., showing that avalanches can arise solely from within cortex and sensitive to excitation-inhibition ratio) is somewhat weak given the wealth of cited in-vitro records showing exactly this (4, 5; 96; 72; 103). I do see how the in-vitro preparation helped calibrating the in vivo methodology (spiking activity and intracellular Ca dynamics), but this is described in the Methods section. Unless I missed something, the authors are encouraged to reconsider the inclusion of this section.

---

## [Author Response]

1) Be clear about the uniqueness of the paper without overstating. That is, two aspects of avalanche dynamics are shown:

(i) A first demonstration of avalanche dynamics linked to a global physiological state (wake vs. anesthetized);

*(ii) A first demonstration of relations between macroscopic avalanche dynamics and spikes, in-vivo. All previous reports relied on population activity measures, not spikes. (A couple of papers by the Chiappalone group, of Genova, did show relations of spiking activity to avalanche dynamics in-vitro, not in-vivo; but these dynamics, due to particular choice of binning in the analysis, were mostly related to activity within synchronized population bursts)*.

We have now re-organized the Abstract. We added a sentence detailing the ambiguity of interpreting macroscopic dynamics using population signals. We have specified the linking of spike avalanches to the global physiological state of wakefulness. We have added our results on spike avalanches in vitro. We have changed our concluding sentence in the Abstract specifically referring to (1) avalanche dynamics are linked to the global physiological state of wakefulness and (2) cortical resting activity organizes as avalanches from local spiking to global populations. We now specifically reference the paper by the Chiappalone group in our Discussion section and point out advances from our in vitro recordings in identified cell types.

*2) Provide a bit more background and context for the work. In other words, explain things (i.e., background theoretical assumptions/limitations etc.) better for the general reader. Specifically, while many citations are provided (e.g., consider the multitude in the first paragraph), there are not straightforward explanations of how neuronal avalanches, power law and criticality are interlinked, and what this actually means in terms of brain functioning. For example, the “being critical of criticality” Beggs and Timme Frontiers 2012 article does a really nice job of these concepts. Perhaps an additional subsection in the Introduction could be written*.

We have now expanded the Introduction by one paragraph and one additional result sentence. Specifically, we have added two new sentences detailing the specific organization of avalanches (scale-invariance) and the emergence of long-range spatial and temporal correlations when being close or at criticality. We also cite the [6] paper together with other reviews. We kept the sentence on improved information transmission to provide the general reader why being critical might be of importance, although we understand that this is not at the center of this work. We have removed the word ‘optimal’ as to not overstate advantages of criticality. In order to provide context for the current study, the final sentence of this paragraph specifies the problem that needs to be overcome for further advances in avalanche research, which is specified for spike avalanches in the next paragraph of the Introduction.

*3) Alternative interpretations should be mentioned and considered, and the potential functional implications of findings should be discussed, via additional paragraph(s) in the Discussion. This should include an explicit discussion on the meaning of the present results in the context of debates that concern whether-or-not actual information processing during tasks operates in a “critical” regime*.

We have now expanded the Discussion in several directions and added subsection headings for better transparency. While this has significantly expanded the Discussion, we believe it has clarified numerous specific and sometimes confusing aspects on avalanche dynamics.

Specifically, we added several subsections referring to methodological and conceptual aspects of the study. We believe these subsections address pitfalls and alternatives in our data interpretation. In light of the complexity of our approach (specific GECI used, resolution of spike activity, how to apply avalanche measures, etc.): A) We added a subsection on the specific GECI used (“Dynamic range and linear estimation of instantaneous firing rates using the GECI YC2.60”). We believe this subsection is important to understand that even the latest 2-PI technology just borders at the threshold to reliably identify avalanches.

B) We added a new subsection on threshold robustness of avalanche dynamics (“Thresholding of continuous firing rate estimates and dependency of avalanche estimates on temporal resolution”). This section addresses the specific relationship between temporal resolution and local firing rate threshold.

C) We added a new subsection on the issue of spatial under-sampling and its predicted effect on a few specific aspects of avalanche parameter estimations, which includes power law fits, cut-offs, and conclusions that should or should not be drawn from ‘spike avalanche’ recordings with microelectrode arrays (“Spatial under sampling of cortical neuronal population and the effect on avalanche parameters”).

Furthermore, we have now re-organized previous subsections on the conceptual advances and implications of our study, which now covers critical states, irregularity, alternative theories, as well as information processing. We believe these subsections better contextualize our findings in other active fields of research in cortical processing. Specifically:

A) We have now streamlined our paragraph on highly irregular activity and neuronal avalanche dynamics and identified two crucial conclusions from our results regarding irregularity and criticality (“Irregular firing with CV>>1, ‘noise’ and neuronal avalanche dynamics”).

B) We added a new subsection on the brain states and avalanche dynamics providing the reader with an overview of the literature on awake state including sleep deprivation and ‘vigilance’ and the contribution of our specific findings (“The awake brain state and avalanche dynamics”).

C) We added a new subsection which discusses recent suggestions that the awake brain might be slightly subcritical or near critical (“Critical dynamics and near critical dynamics”). This section specifically addresses current, important discussions within the criticality community.

D) We added a new subsection which points out and discusses the significance of our in vitro results (“In vitro experiments establish additional benchmarks of avalanches in cortical layers”).

Currently, there are no experimental results published that would answer the question “whether-or-not actual information processing during tasks operates in a “critical” regime”. We agree with the editors that this is an extremely important and pressing topic. Unfortunately, there is only one computational study showing that marginally subcritical dynamics explains enhanced stimulus discriminability under attention (111) (cited in our section “Critical dynamics and near critical dynamics”). In addition, although we have currently unpublished results demonstrating that in the behaving macaque monkey avalanches dynamics is maintained during a visual discrimination task (Shan et al. 2014 SfN abstract), we consider these results too preliminary to justify a whole section. We specified though in our ‘final conclusions’ the potential advantages of critical layer 2/3 networks with respect to processing of layer 4 inputs.

*4) Overstatements should be removed. For example, in Abstract, reduce the tone where avalanche dynamics are attributed “optimal” for information processing. The results do not relate to “function”, but rather to an awake state. Assuming that other types of neurons (or pyr cells at other layers) have not been measured, do not over-state the relation of avalanche dynamics to pyramidal cells*.

In the Abstract, we have now removed the attribute “optimal” and “information processing” from our conclusion sentence.

Regarding potentially overstating the relation of avalanche dynamics to pyramidal neurons, we apologize for this misunderstanding. While our method for achieving GECI expression (in utero electroporation) specifically targets pyramidal neurons, we never intended to project the idea that layer 2/3 pyramidal neurons are the only cell type required for avalanche generation. We believe though that it is important to demonstrate avalanche organization in pyramidal neuron group output. First, interneurons due to their high firing rates can in principle inflate spike clusters, in which case large cluster sizes might simply reflect neuronal cell type differences rather than interactions within neuronal groups. Second, by focusing on pyramidal neurons, we are studying the computational output of a local cortical microcircuit. We now state clearly that we do not assume our data show that avalanches only require pyramidal neurons. These three aspects are stated upfront at the beginning of our Discussion and are also revisited at other places in the text. We have also changed the title accordingly by replacing ‘emergence’ with the more neutral statement of ‘organization’. Our results do not exclude the possibility that signatures of criticality might be found in other cell types.

*Reviewer #2*:

*1) The manuscript “Avalanche dynamics emerge from irregular spiking of pyramidal neurons in the awake resting state” by Bellay et al. demonstrates that the spatial and temporal correlation of (suprathreshold) firing events in superficial layer pyramidal cells can be described as avalanches in the awake state to a larger degree than during anaesthetized and awaking states. The authors first show that their estimate of instantaneous firing (λ) from calcium signals is linked to action potental firing (*Figure 1*) and they show that neurons fired irregular with long periods of quiescents in all states and showed largest variabilitiy (CV) during the awakening period (*Figure 2*). More importantly, the study reports that neural firing (indexed as λ) showed only low correlations with somewhat more correlated firing during awakening that during the awake state. Interestingly, avalanche dynamics was more pronounced during the awake state (*Figure 5—figure supplement 1*, with the so called avalanche life time distributions following a power law as described before). The authors also show in in vitro recordings that the temporal correlations are reduced (with inhibition) or enhanced (during disinhibition) but remained intact during moderate reductions of GABAa, AMPA or NMDA synaptic activity (*Figure 7*). Spatial clustering was absent in these in vitro recordings*.

The manuscript convincingly demonstrates avalanche dynamics in superficial layer pyramidal cell firing that is state dependent with enhanced avalanche dynamics during the awake state. This insight follows from an extensive characterization for the temporal and spatial clustering of firing events from in-vivo calcium recordings. Methods, experimental approach are all state-of-the-art.

We thank the referee for her/his favorable evaluation of our study, in which we utilized the current potential of very sensitive genetically-encoded calcium indicators, the linkage between the anesthetized and awake state, and the conceptual approach to avalanche analysis. Towards a better understanding of these various aspects of our work and in response to the general editorial comments, we have now expanded the Discussion to address these issues in much more detail.

*However, the manuscript did not attempt to quantify why or how the avalanche dynamics is influencing information processing in the circuits. As such the study provides more of a proof of principle, using sophisticated tools, that temporal and spatial clustering's are evident in calcium imaged firing rate dynamics of large populations of neurons (as it is evident in all population metrics studied so far to my knowledge)*.

We agree with this referee that temporal and spatial clustering is evident in most imaging studies that capture firing activity in large populations of neurons. The unique result in our study, though, refers to the finding that in the awake state the clustering attains a scale-invariant organization as well as other hallmarks of neuronal avalanche dynamics. The manuscript in its present form thus establishes a proof of principle for critical resting activity, which lays the groundwork for future studies of avalanche dynamics at the cellular level, including during active information processing.

We have now moved the sentences on avalanches, noise correlations and evoked responses from the Introduction to the end of the Discussion (in the subsection headed “Final conclusions”). We have changed the previous statement on optimal information processing to a general sentence on existing work about avalanches and improved information processing in the Introduction (second paragraph). We believe it is important to provide the general reader with some reasoning why one should care about these dynamics in the long term.

2) If possible, the study would increase its appeal by showing how (in quantitative terms) e.g. a stimulus pulse, (or maybe a spontaneous bursting event) directly predicts some events/dynamics at later time points (or whether it predicts the awake state). Such an approach would help to identify whether there are actual information processing effects resulting from events that are characterized by avalanche dynamics.

It is not clear whether avalanche dynamics also govern evoked activity or activity during behavior. We are currently concluding a study in non-human primates using high-density microelectrode arrays to study avalanche dynamics during behavior. However, the non-stationary activity during behavior poses significant challenges for this type of analysis and is still in progress ([121] SfN). We believe the treatment of stimulus responses would go beyond the scope of the current work, but it is clearly a part of our future plans.

We have now performed the analysis suggested by the referee, in which spontaneous, large avalanches are used as trigger to compute post-event histograms (a similar approach was taken in (Plenz and Chialvo, 2009; [74])). The corresponding figure is (Figure 8) for the awake state at two different temporal resolutions (240 ms, black, 88 ms blue) and the anesthetized state (250 ms). These distributions show that for all three conditions, activity rapidly declines beyond ∼1s. Further analysis using larger neuronal populations and longer recordings might be required to identify general features for these histograms. Given the uncertainty in interpretation, we prefer not to include these results in the current manuscript.

Author response image 1.Post-event time histogram of cluster occurrence after occurrence of a large cluster.In general, large clusters are followed by cluster activity for about 10 s (average over all recordings for each state). Temporal resolution: 1x = 250 ms; 4x ∼88 ms. No significant differences were observed for clusters activity following large clusters between the awake or the anesthetized state.**DOI:**
http://dx.doi.org/10.7554/eLife.07224.020

*3) There are some aspects of the manuscript that deserve consideration and which are listed here*:

The Title needs some clarification. In the text authors mention that avalanches depend on balance of excitation and inhibition. Since inhibitory neurons are not identified in the current work but seem to play a role, it seems a foregone conclusion to state that it is the firing of pyramidal cells that lead to avalanches as is suggested by the Title. Do the authors imply that avalanche dynamics would not be evident when recording interneuron activation patterns?

We apologize for this misunderstanding. We never intended to imply that pyramidal neurons alone suffice to establish avalanche dynamics. In fact, our excitation/inhibition experiments in vitro clearly demonstrate that inhibitory circuits are also involved. We have changed the Title accordingly to make this distinction and re-examined the text. Please also see our response to Editorial point #4. The referee is absolutely correct that certain interneuron populations might equally well display avalanche signatures, but we do not have conclusive experimental evidence at this point.

*As far as I see the authors infer that distributions follow scale free power law behavior simply from the fact that the fits are significant and that shuffling of the data do not result in significant fits. While this may be the standard in the field I am worried that there are no alternative hypotheses offered or tested. Shouldn't a statistical test compare two functions in their degree of predicting the distribution? I ask this because it is one thing to describe the dynamics of the system as satisfying a scale free power law, but the interpretation and inferences would be different if other non-scale free functions describe the data in similar good ways. What would happen to the criticality perspective if future studies would show that the dynamics would follow only “somewhat” a power law, but that the fine scale dynamics can be as well described by scale specific function? This is probably not only unclear to ‘non-technical’ readers*.

We agree with the referee on his conceptual and methodological considerations. In fact, our statistical analysis supporting power law scaling of avalanche size distributions compares the power law hypothesis to the alternative hypothesis of an exponential distribution as expected from independent firing (and as shown for our shuffled data). Specifically, we use the log-likelihood ratio (LLR) test, which compares these two functions in their goodness of fitting the empirical distribution. For clarification, we have added this information to the quantification of alpha in the Results section (in the subsection headed “Avalanches emerge from awake neuronal firing”).

Regarding the comparison with other heavy-tail distributions than a power law, we would like to refer to recent studies from of our lab where we did exactly that. In (41), we compared the power law model to other heavy-tail distributions for LFP based avalanche data and demonstrated that the power law is the most favored model for these size distribution up to the cut-off, which we recently demonstrate to contain distinct under-sampled patterns and should not be included in the fit (121). Based on these published analyses, in the current study, we identified the cut-off and focused our state comparisons (awake, waking, anaesthetized) on the power law model.

We would like to point out that our conclusions in support of avalanche dynamics are not only based on the power law fit. We show additional and partly independent measures expected for avalanche dynamics (branching ratio sigma, lifetime distribution, threshold independence, scaling of exponents for cluster size and lifetime distributions). We now address these and related questions in the newly added subsection “Critical dynamics and near critical dynamics”.

*The authors estimate the relation of the inferred λ-firing with actual firing rates measured with a Neuronexus array (in the subsection headed “Analysis of calcium transients from surrogate data from different temporal resolutions”). It is crucial to add information on how likely this array records neurons from similar layers as in the calcium measurements. Are firing rates from L2/3 different between layers? What is the firing rate in VTA neurons and their relation to λ. Some more explicit information on this would be helpful*.

We have now provided additional information about the array configuration, array insertion and that we also recorded nLFP based avalanches during these experiments which are primarily found in superficial layers (in the subsection headed “Analysis of calcium transients from surrogate data for different temporal resolutions”).

We currently do not know whether firing rates of pyramidal neurons are different between layer 2 and layer 3. Our 2-photon imaging experiments were not designed to discriminate between these layers and the Neuronexus arrays have an interelectrode distance of 200 µm, which is too coarse to make this discrimination.

Regarding the firing rate in VTA neurons and their relation to λ, see Figure 9, which demonstrates experimentally that avalanche dynamics are not affected in cortex-VTA co-cultures when the VTA or midbrain culture is acutely disconnected from the cortex. This work was done in the context of a developmental study using cortex-VTA cultures as an in vitro model (28).

Author response image 2.Acutely disconnecting of the VTA does not affect neuronal avalanche dynamics in the cortical part of the co-culture.(a) Light microscopic picture of a cortex (Ctx)-Ventral Tegmental Area (VTA) co-culture grown on a planar microelectrode array (MEA). Left: before acute dissection. Right: after acute dissection of the connection between the two tissue regions (broken line). (b) Spontaneous gamma-burst in the LFP from a single electrode in the cortex part of the culture before and after the dissection. (c) Cluster size distribution before and after dissection (average from 3 cultures). Note that in this plot the probability of cluster size s, i.e. P, is multiplied by s^alpha which translates the power law into a horizontal graph up to the cut-off.**DOI:**
http://dx.doi.org/10.7554/eLife.07224.021

*Reviewer #3*:

*The concept of neural avalanche dynamics was introduced by Beggs and Plenz over a decade ago, and continues to serve as a fruitful metaphor in analyses of experimental data and in development of theory. It is an appealing concept from both the mechanistic and functional aspects: providing clues for the origin of scale invariance in the statistics of neural network activity, as well as offering a point of view on the enrichment of network activity patterns by means of matched excitation-inhibition ratios*.

In this paper Belly et al. provide convincing evidence for the occurrence of avalanche dynamics—scale invariant cascades of neuronal spiking activities—in the rat superficial cortical layers. Moreover, they show that these dynamics are affected by a global state of the animal—anesthetized vs. awake—and provide a nice link between single neuron irregular firing statistics and population statistics.

We thank this reviewer for her/his favorable evaluation of our paper demonstrating for the first time the link between single neuron irregular firing statistics and avalanche population statistics in vivo.

*I have one main comment that concerns paper's structure. I do not understand why the authors include in this paper the in-vitro results. They seem irrelevant to the point made, and even the authors themselves do not mention these results in the Abstract, nor discuss these data in the Discussion section. The only mentioned justification for the choice (i.e., showing that avalanches can arise solely from within cortex and sensitive to excitation-inhibition ratio) is somewhat weak given the wealth of cited in-vitro records showing exactly this (*[4]*,*
[5]*;*
[96]*;*
[72]*;*
[103]*). I do see how the in-vitro preparation helped calibrating the in-vivo methodology (spiking activity and intracellular Ca dynamics), but this is described in the Methods section. Unless I missed something, the authors are encouraged to reconsider the inclusion of this section*.

In our initial submission, we indeed removed many of the original arguments for our in vitro results simply for space reasons. However, our in vitro results are of considerable fundamental value for the following two main reasons: (1) Previous in vitro studies suffer from the ambiguity of LFP population signals and the cellular identities in extracellular unit recordings. For example, strongly bursting interneurons can lead to large spike clusters, in which case heavy-tail size distributions are more likely to reflect cell type diversity rather than neuronal interactions. (2) Our in vitro results also demonstrate that using a high-threshold GECI still allows the identification of neuronal avalanche dynamics. This proof could not be conducted in vivo because the high threshold reduces the number of events recorded, requiring long recording times that are currently not easily attainable in the awake animal.

By extending our experimental proofs to the in vitro case, we are opening up the field of criticality research to take advantage of the in vitro experimental designs. This will be of particular importance for future studies that try to identify mechanisms of avalanche regulation in cortical tissue, which can be much easier studied in vitro than in vivo. We have now explicitly stated the limitations of previous in vitro studies in the Results section (in the subsection headed “AP avalanches in L2/3 PN in vitro depend on the E/I balance”). We have added our in vitro results to the Abstract and added a section in the Discussion specifically related to our in vitro findings (“In vitro results experimentally establish threshold-invariant avalanche dynamics in L2/3 PN regulated by the E/I balance”). We hope that this referee is supportive of the argument that in vitro studies can provide fundamental insights into mechanistic aspects of cortical dynamics. We believe that our in vitro results provide an important complementary proof to our in vivo findings and should not be removed from the manuscript.